**Subject Category:**
Biology (whole organism)

palaeontology

fossil record, pinnipeds, marine mammals, historiographical revision, paleobiology database, biases in the fossil record

**Author for correspondence:**
Ana Valenzuela-Toro
e-mail: avalenzuela.toro@gmail.com

# What do we know about the fossil record of pinnipeds? A historiographical investigation

Ana Valenzuela-Toro[1] and Nicholas D. Pyenson[2,3]

[1]Department of Ecology and Evolutionary Biology, University of California Santa Cruz, Coastal Biology Building, 130 McAllister Way, Santa Cruz, CA 95060, USA
[2]Department of Paleobiology, National Museum of Natural History, Smithsonian Institution, PO Box 37012, Washington, DC 20013, USA
[3]Department of Paleontology and Geology, Burke Museum of Natural History and Culture, Seattle, WA 98105, USA

AV-T, 0000-0003-1497-364X; NDP, 0000-0003-4678-5782

The fossil record of pinnipeds (seals, fur seals and walruses) is globally distributed, spanning from the late Oligocene to the Holocene. This record shows a complex evolutionary history that could not otherwise be inferred from their extant relatives, including multiple radiations and iterative ecomorphological specializations among different lineages, many of which are extinct. The fossil record of pinnipeds is not uniformly represented in space and time, however, leaving some gaps in our knowledge. We performed a historiographical investigation of the published fossil record of pinnipeds based on the information available in the Paleobiology Database, with the aim to broadly characterize and evaluate it from a taxonomic, geographical and temporal perspective. We identified major trends, strengths and weaknesses of the pinniped fossil record, including potential biases that may affect our interpretations. We found that 39% of the record corresponds to extant taxa, which are essentially from the Pleistocene and Holocene. There is a larger record from the Northern Hemisphere, suggesting biases in sampling and collection effort. The record is not strongly biased by sedimentary outcrop bias. Specifically, for extinct species, nearly half of them are represented by a single occurrence and a large proportion have type specimens consisting of single isolated postcranial elements. While the pinniped fossil record may have adequate temporal and taxonomic coverage, it has a strong geographical bias and its comparability is hindered by the incompleteness of type specimens. These results should be taken into account when addressing patterns of their past diversity, evolutionary history and paleoecology.

# 1. Introduction

Among living marine mammals, pinnipeds constitute the second most species-rich clade, comprising 33 widely distributed extant species in three subclades: Phocidae (true seals, with 19 species), Otariidae (fur seals and sea lions, with 14 species) and Odobenidae with only one extant species, walruses (*Odobenus rosmarus*; [1,2]). The fossil record of pinnipeds is based on localities from the late Oligocene to the early Miocene of the North Pacific Ocean and from the early Miocene of the Mediterranean and Paratethys regions, through the Holocene from both the Northern and Southern hemispheres. From the fossil evidence, it is clear that pinnipeds were different in the geologic past, showing greater species richness in some groups, different ecomorphologies, body sizes and geographical distributions [2]. For instance, the fossil record shows that, during the Neogene, Odobenidae comprised at least 20 species (versus the monotypic extant walrus) with a range of body sizes and skull morphologies that imply much greater morphological disparity ([2–4]; and references therein). Faunal comparisons suggest that the rise and fall of different pinniped subclades are likely dependent on changing oceanographic and habitat availability (e.g. haul outs for rookeries) over geologic time [5–7].

As with other marine mammal groups, the fossil record of pinnipeds appears to possess certain qualities, modes and patterns. For example, despite their relative abundance in fossiliferous sites throughout the world, fossil pinnipeds are not uniformly described across the world (see [2]). Furthermore, the fossil record of some groups is represented by isolated and fragmentary remains (e.g. [8,9]). These attributes may bias the fossil record of pinnipeds in a similar way to the fossil record of cetaceans and sirenians, which show biases in geography, preservation and historiography [10,11]. Hence, it remains unclear how potential sources of bias, such as geographical and temporal occurrences, or sampling effort, affect our knowledge and interpretations of the fossil record of pinnipeds.

Here, we perform a historiographical analysis of the published fossil record of pinnipeds using the Paleobiology Database with the aim to qualitatively and quantitatively characterize the record from taxonomic, geographical and temporal perspectives, identifying the principal trends, strengths and weaknesses, as well as to explore the potential biases that may affect our interpretations about the past diversity, evolutionary history and paleoecology of this group of marine mammals.

# 2. Methods

## 2.1. Data collection

Raw data were downloaded from the Paleobiology Database (PaleoDB; available at http://www.paleobiodb.org) on 29 April 2019, using the name 'pinnipedimorpha'. We restricted the search to output the files 'fossil occurrences' and 'valid taxa' for our analysis and also to provide a more detailed analysis of the type material of valid extinct species of pinnipeds. In this study, we regard the term 'occurrences' to signify the records of a taxon that are grouped into geographical collections, logged into the PaleoDB. Furthermore, for our study, we only examined published (i.e. documented) occurrences, where fossil remains have been taxonomically identified and also belong to a discrete geographical collection with precise coordinates stipulated in the scientific literature. Thus, we excluded, to the best of our ability, fossil occurrences of pinnipeds that have not been formally published in the peer-reviewed literature; in cases where the PaleoDB's occurrence data (i.e. geographical, stratigraphic or specimen-related) can be more precisely ascertained, we have noted the supplemental information. Because our analysis emphasizes published occurrences reflected in museum collections, we also note that there are additional unpublished fossil occurrences worth further discussion (see more detail below).

For further information about definitions used by the PaleoDB, see Peters & McClennen [12]. Their revision includes fossil (older than 11 000 years old) and subfossil (younger than 11 000 years old) occurrences; however, we refer to them indistinctly as fossils. The raw data were complemented by information from different sources, including the PaleoDB website and peer-reviewed literature.

## 2.2. Categorical analysis

To characterize the taxonomic identity, the temporal and geographical distributions, and the historiography of fossil pinniped publications, we included the following categories: taxon name, taxonomic rank, geographical provenance (hemisphere, oceanic basin and country), geological formation/locality, geological age, primary language of the publication, year of the publication and full reference (see

electronic supplementary material, table S1). When it was not possible to obtain information, we recorded 'No information' in the respective category.

For the taxonomic classification, we used the following groups, as listed in the PaleoDB: enaliarctids, which include basal pinnipedimorphs such as *Enaliarctos* and its relatives (including *Pinnarctidion* and *Pteronarctos*); desmatophocids, which include species belonging to the extinct genera *Allodesmus* and *Desmatophoca*; Pan-Otariidae, which include species belonging to the extinct genus *Eotaria*; and the extant families Phocidae, Otariidae and Odobenidae. For the 'oceanic basins' category, we followed a modified version of Berta *et al*. [13] and Deméré *et al*. [14] as follows: eastern and western North and South Pacific Ocean, eastern and western North and South Atlantic Ocean, Parathethys region, Indian Ocean, Lake Baikal, North Sea, Baltic Sea, Mediterranean Sea, Southern Ocean and Arctic Ocean. In particular, the Paratethys region referred to the occurrences placed in continental Europe and other regions considered remnants of the ancient Neo-Parathethys Sea (e.g. Black and Caspian seas, Austria, Czech Republic, Hungary, Romania, Serbia, Bulgaria, Moldavia, Ukraine, Russia, Georgia, Azerbaijan, Iran and Kazakhstan; see [15]). The Mediterranean Sea comprises occurrences from the adjacent areas to this sea and also northern Africa.

## 2.3. Analysis of rock outcrop area and fossil occurrences

We explored the association between sedimentary rock outcrop area from the Neogene (measured in $km^2$) of the USA and the number of reported occurrences of fossil pinnipeds from this area, using unpublished marine rock area data collected by Uhen & Pyenson [10] in 2007.

## 2.4. Museum collections

We examined the relationship between the number of fossil specimens from two most fossil-rich geologic units in terms of the abundance of extinct pinniped remains in the USA: Calvert and Yorktown formations and the number of published fossil occurrences from those units. For that, we focused on the pinniped collection from those units housed in the Department of Paleobiology at the Smithsonian Institution's National Museum of Natural History in Washington, DC, USA (data downloaded at https://collections.nmnh.si.edu on 2 October 2018).

## 2.5. Collection curves

Collection curves from the documented occurrences of extinct species data were generated using the software Past (v. 3.23; [16]). In Past, we selected the function individual rarefaction by estimating the number of taxa (extinct species) expected to be collected in a sample with a small initial total number of individuals. The input data file for Past contained occurrences of fossil species for a given geological formation that were grouped by year. For this analysis, we considered the Calvert, Astoria, Yorktown and Purisima Formation, given the number of taxa reported in each one.

## 2.6. Analysis of extinct species

To identify the principal trends for type specimens of extinct species, the following categories were included: the catalogue number (= collection number within a research institution) of the type specimen, type locality, skeletal element(s) of the type material (skull, cranium, skeleton or postcranium) and their articulation and completeness conditions. For the analysis of the skeletal elements, we followed White & Folkens [17] in defining skull as the entire bony structure of the head (dermatocranium, splanchnocranium and neurocranium) including the lower jaw (or mandible), differing from the terminology cranium, which only corresponds to the skull without the lower jaw. We used a modified version of Pyenson *et al*. [18] and Boessenecker *et al*. [19] for the articulation and element association of type material, defined here as: category 1, complete or almost complete articulated skeleton; category 2, disarticulated skeletons and associated skeletal elements, including cranial and other skeletal elements; and category 3, isolated elements.

Caribbean monk seals (*Neomonachus tropicalis*) and Japanese sea lions (*Zalophus japonicus* according to [20]; or *Z. californianus japonicus* according to [21]) became extinct over the last century; for the purpose of the paper, these species were categorized as extant [22].

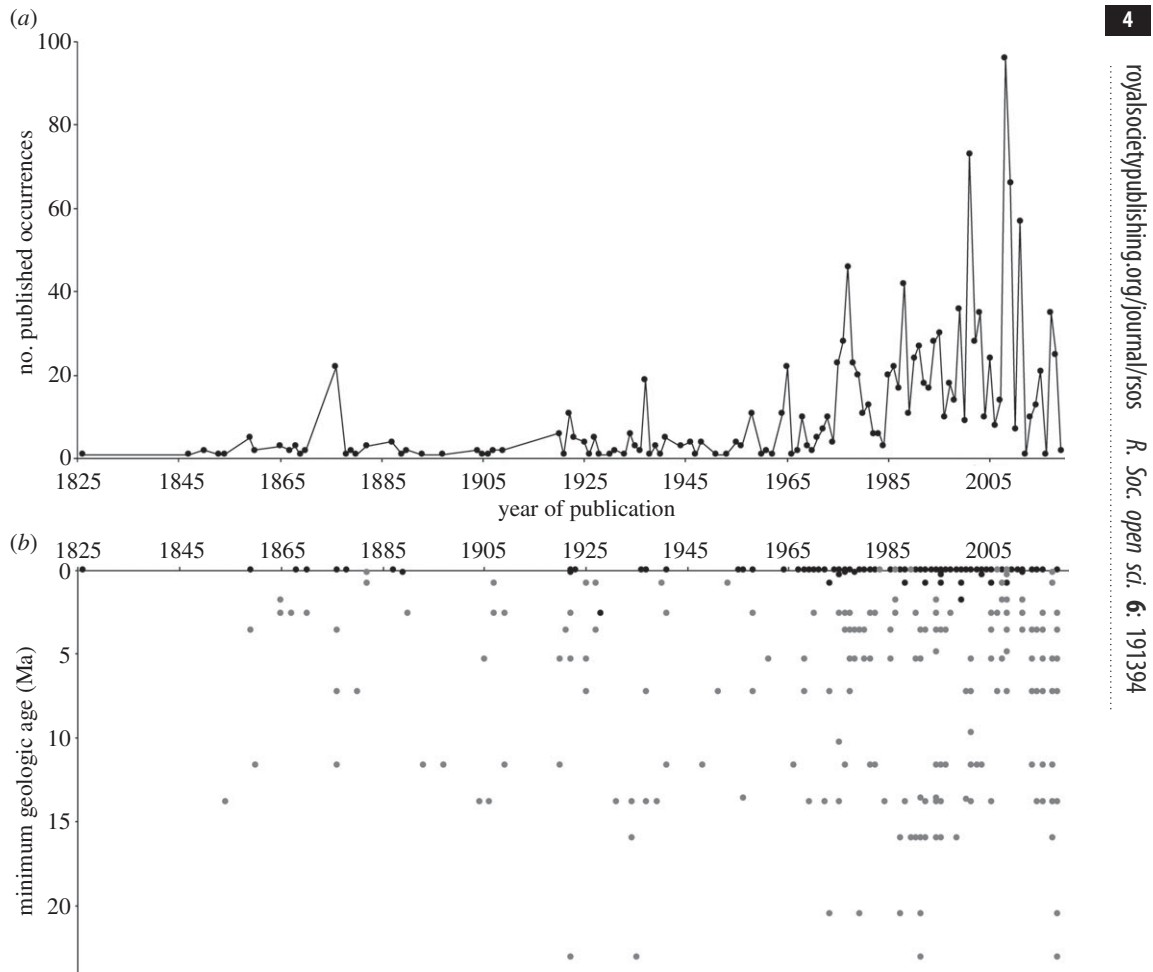

**Figure 1.** Published fossil occurrences over time. (*a*) Number of publications of fossil pinnipeds through calendar years (*n* = 1296). (*b*) Minimum geological age of fossil occurrences (in millions of years; Ma) identified at level of species through calendar years. Black dots represent occurrences of extant species and grey dots represent occurrences of extinct species.

## 2.7. Zooarchaeological record

Finally, a note of caution: the zooarchaeological record of pinnipeds is extensive (see [23–33]; and references therein); however, these data are not consistently entered into the PaleoDB. In addition, some of the Holocene occurrences listed in the PaleoDB are modern findings (e.g. living taxa) rather than zooarchaeological or subfossil remains (M. Uhen, personal communication). Thus, we performed a more in-depth verification of these reports, excluding improper zooarchaeological occurrences. Nevertheless, the addition of more comprehensive zooarchaeological data may potentially alter the results presented herein.

# 3. Results

## 3.1. Documented fossil record

The compilation resulted in 1296 occurrences of fossil pinnipeds identified at a variety of taxonomical ranks (i.e. superfamily, family, subfamily, genus, subgenus, species, subspecies, tribe and subtribe) and published in the peer-reviewed literature. The number of publications reporting fossil pinnipeds, including extinct and extant taxa, fluctuates through calendar time. There is, however, a general increase in the number of publications reporting new occurrences over the last 3 decades (figure 1*a*). The oldest publication in our survey dates to 1826, with the report of fossil remains of extant harbour

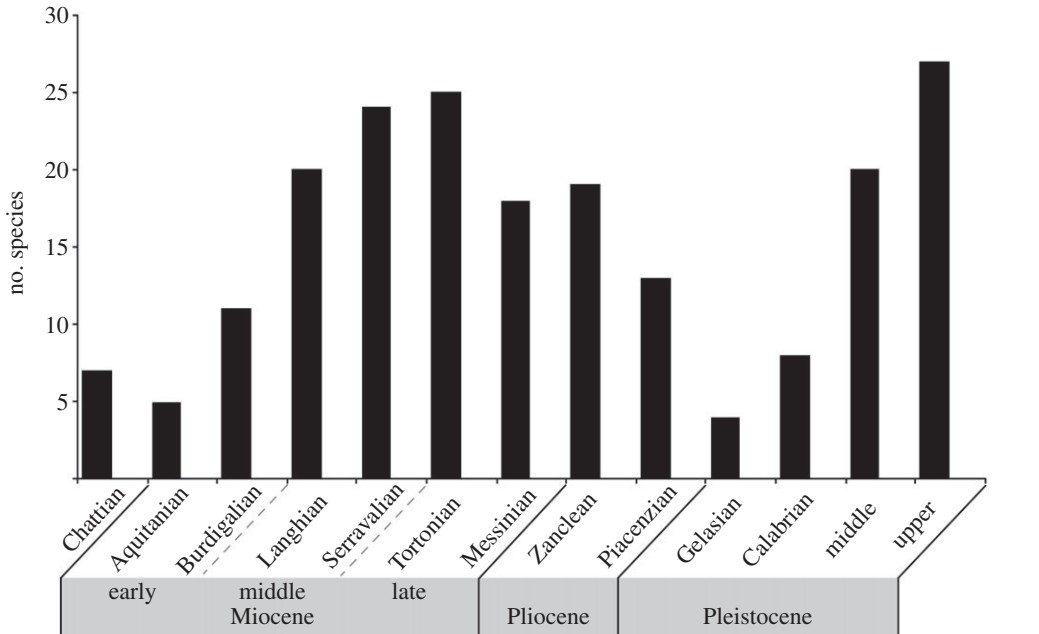

**Figure 2.** Number of species (both extant and extinct) over geological time. Note that the Chattian Stage belongs to the late Oligocene.

seals (*Phoca vitulina*) from the Pleistocene of the North Sea [34]. This record is followed by the occurrence of an indeterminate phocid from Calvert Formation (middle Miocene) of the eastern coast of North America. More than half of the pinniped fossil record (58%; $n = 752$) has been reported since 1990.

Remarkably, 90% ($n = 1160$) of the published occurrences (including all taxonomic ranks and ages) are derived from the Northern Hemisphere. The most productive geographical sub-regions in terms of occurrences are the eastern North Pacific (29%; $n = 374$), western North Atlantic (17%; $n = 215$) and North Sea (13%; $n = 162$). Occurrences from the Arctic, western North Pacific and the Paratethys Region represent 9% ($n = 120$), 8% ($n = 103$) and 7% ($n = 92$), respectively. Fossil occurrences from other regions (including from the Southern Hemisphere) are minor; the eastern coast of the South Atlantic Ocean, eastern coast of the South Pacific Ocean, the western coast of the South Pacific Ocean and the western coast of the South Atlantic Ocean represent only 5% ($n = 59$), 4% ($n = 47$), 2% ($n = 21$) and 1% ($n = 8$), respectively, of published fossil pinniped occurrences.

The distribution of the published occurrences over geologic time spans from the late Oligocene (approx. 26 Ma) to the Holocene (figure 1*b*; electronic supplementary material, figure S1). Additionally, there are a few putative older occurrences of fossil pinnipeds ($n = 2$) from the early Oligocene [35,36]. However, the validity of these occurrences and their stratigraphic provenance remain controversial (see [14,37]). The Quaternary (Pleistocene and Holocene) yielded the largest amount of fossil occurrences, encompassing 51% ($n = 665$) of the record (see below). Occurrences from the Pliocene represent 14% ($n = 187$). Reports from the Miocene–Pliocene and Miocene represent 33% ($n = 430$). Occurrences from the late Oligocene represent 1% ($n = 14$). It is worth noting that those differences are possibly correlated with the extension of the time span comprising the temporal bins used, and not necessarily implying particular properties during those temporal windows.

The number of species of both extinct and extant pinnipeds over geological time (at level of stage) is variable with peaks during the Serravallian and Tortonian and the Pleistocene (figure 2). Furthermore, to assess the effect of the availability of pre-Quaternary sedimentary rocks over the number of fossil pinnipeds, we plotted pinniped richness data of extinct and extant taxa against rock outcrop area for marine rock units of the western and eastern coasts of the USA (figure 3). These rocks comprise one of the more consistently sampled areas for fossil pinnipeds and other fossil marine mammals in the world. We used unpublished marine rock area data collected for similar analyses with fossil cetaceans reported in Uhen & Pyenson [10]. These geological data range at the subepoch level from the early Eocene through Pliocene of North America, following Uhen & Pyenson [10]'s data. Middle Eocene rocks, especially from the southeastern US coastal plain, account for the bulk area of Cenozoic rock outcrop in North America, but even excluding rocks of this age from our analysis (because of the lack of Eocene age pinnipeds in this continent), we did not find a correlation between the map area of

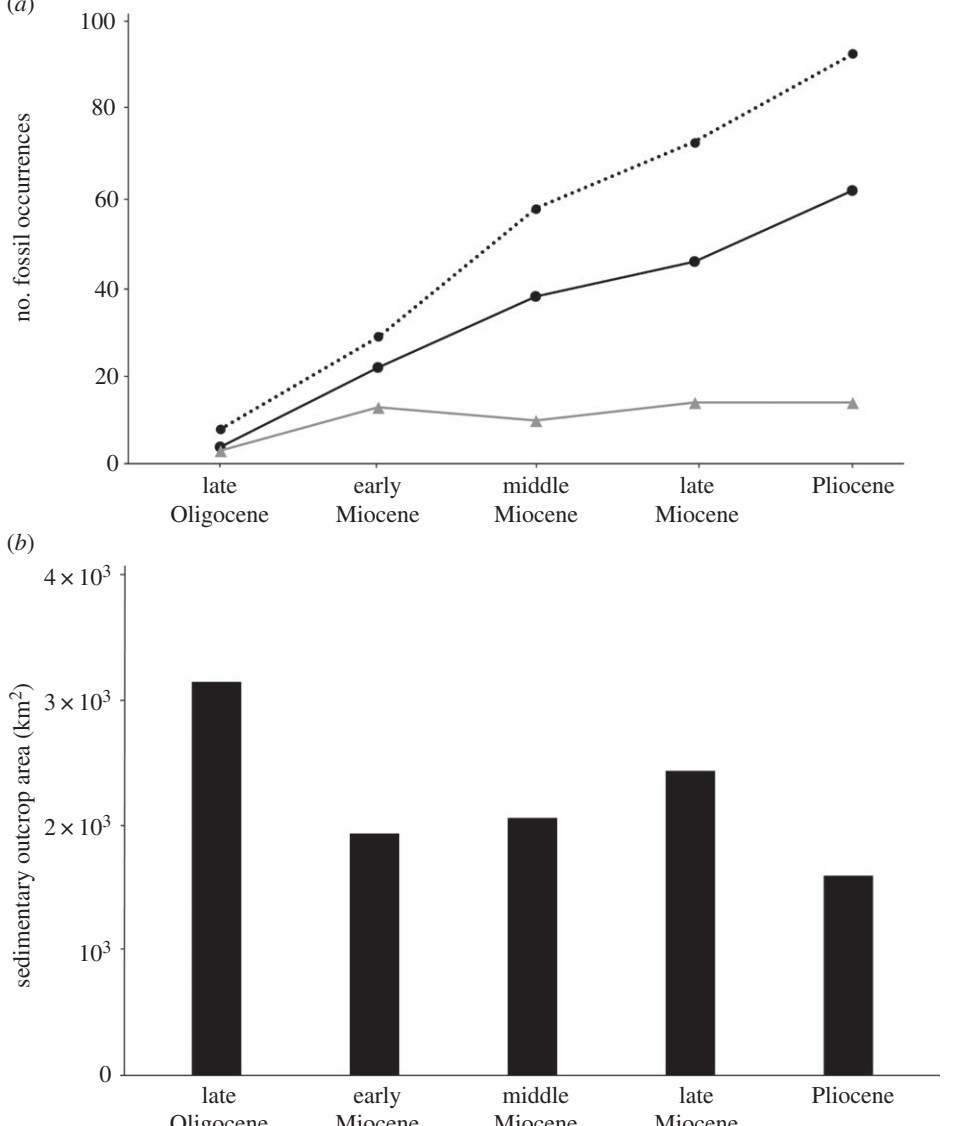

**Figure 3.** (a) Number of pinniped occurrences from the eastern and western coasts of the USA through geological time. Dashed line represents the number of fossil occurrences including an assortment of taxonomic ranks. Black line represents the number of fossil occurrences identified at the level of species. Grey line represents the species richness of pinnipeds. (b) Rock outcrop area for North American marine rock units, using unpublished data collected for the analyses published in Uhen & Pyenson [10].

rocks from USA and the number of published occurrences of pinnipeds through geological time (figure 3). Furthermore, we did not find a correspondence between the species richness over geological time and the sedimentary map area. For this analysis, we excluded putative occurrences of pinnipeds older than the late Oligocene and two occurrences identified as phocids from the late Oligocene because their validity is controversial (see above). It is worth noting that strong differences between rock area along the eastern and western coasts of the USA do exist: for example, abundant early Miocene rocks are known from California and Oregon, whereas similarly aged rocks are far less abundant (by areal map data) along the east coast of the USA.

## 3.2. Fossil record of extant pinnipeds

Of the 1296 documented fossil occurrences of pinnipeds, 37% ($n = 480$) correspond to fossil and zooarchaeological occurrences of extant species. Species belonging to Phocidae are the most abundant representing 48% ($n = 231$) of record of extant species, which is followed by members of Otariidae (29%; $n = 141$) and Odobenidae (23%; $n = 108$). The extant species with larger representation in the fossil record are *O. rosmarus* (walruses), *Arctocephalus townsendi* (Guadalupe fur seals), *Pusa hispida*

(ringed seals), *Erignathus barbatus* (bearded seals), *Phoca vitulina* (harbour seals) and *Mirounga angustirostris* (Northern elephant seals).

Almost all (99%; $n = 478$) of the fossil and zooarchaeological occurrences of extant species are referred to occurrences from the Pleistocene and the Holocene epoch (electronic supplementary material, figure S1). The only exceptions are constituted by a single occurrence of *O. rosmarus* from the Miocene of Belgium [38]; and remains assigned to *Eumetopias jubata* (Steller sea lions) from the Pliocene of Japan [39]. A third record, corresponding to remains of *Phocarctos hookeri* (New Zealand sea lions) from the Pliocene of New Zealand [40] is included in PBDB; however, it has been argued that this record corresponds to a Holocene specimen, younger than 1000 years old, rather than a Pliocene occurrence [41].

## 3.3. Fossil record of extinct pinnipeds

A total of 388 occurrences of extinct species belonged to one of the 102 valid extinct species (electronic supplementary material, tables S1 and S2). We observed a positive trend of growing publications on extinct species over calendar years (figure 4a). In particular, 1977, 2001 and 2008 represent outstanding years, correlating with the publication of more than 30 occurrences of extinct species in each calendar year in three seminal and monographic publications ([8,42,43]; figure 3a). Furthermore, 61% ($n = 235$) of the occurrences of extinct species have been reported since 1990. Of the 102 species, only five extinct species belong to an extant genus (*Callorhinus gilmorei*, *Histriophoca alekseevi*, *Neophoca palatina*, *Otaria fisheri* and *Phoca moori*), representing approximately 5% of the extinct species.

The fossil record of extinct species spans from the late Oligocene to the late Pleistocene (figure 1). When analysed by geologic stages, we found there is normal-like distribution of occurrences over the Neogene and Quaternary (electronic supplementary material, figure S1) with a peak of known occurrences from the Serravallian, Tortonian, Zanclean and Piacenzian, comprising together 63% ($n = 246$) of the record. More recent occurrences of extinct species from the Pleistocene (including the Gelasian, Calabrian and Middle and Upper Pleistocene) represent 8% ($n = 30$) of the record.

In terms of their geographical origin, 93% ($n = 362$) of the record of extinct species is from the Northern Hemisphere, which is practically identical to the results found for the all published occurrences (including extant and extinct taxa, see above). North America concentrates more than half of the record, followed by Europe, Asia, Africa and South America (electronic supplementary material, figure S2). Six North American sedimentary formations lead the list of the most prolific pinniped-producing units, in terms of the number of published occurrences of extinct pinnipeds (identified at level of species), in decreasing rank: Calvert (eastern USA), Yorktown (eastern USA), Purisima (western USA), Santa Margarita (western USA), Astoria (western USA) and Bone Valley (eastern USA) formations. In fact, the Calvert and Yorktown formations (Middle Miocene and late Miocene-early Pliocene in age, respectively) comprise 6% ($n = 25$), each of the published occurrences of extinct pinniped species. Surprisingly, the Pisco Formation of Peru represents the seventh most prolific unit comprising 2% ($n = 9$) of the fossil occurrences of extinct taxa. Unfortunately, it was not possible to determine a clear and unambiguous geological formation for 11% ($n = 42$) of the occurrences of extinct taxa. The taxonomic accumulation curves (rarefaction curves; electronic supplementary material, figure S3) show that neither Calvert, Astoria, Yorktown and Purisima formations have reached a saturation in sampling over time. In turn, this analysis suggests that more fossil pinniped species should be found from these rock units.

Using data from museum collections of the two most fossil-rich units (Calvert and Yorktown formations; electronic supplementary material, tables S4 and S5), we investigated whether there is a correspondence between the number of pinniped specimens in these collections and the published record from those units. We discovered that the museum collections for the Calvert Formation comprise a disproportionately smaller number of specimens ($n = 125$) compared with the number of specimens from Yorktown Formation ($n = 4153$), which contrasts with the published fossil record from both units ($n = 25$ from Calvert Formation and $n = 21$ from Yorktown Formation). When examining the taxonomic richness of Calvert and Yorktown formations, we obtained dissimilar results. Despite having the highest number of published fossil occurrences with 25 reports of extinct taxa, only 2 species of phocids have been identified from the Calvert Formation: *Leptophoca proxima* and *Monotherium? wymani*. On the other hand, six species of pinnipeds have been reported from the Yorktown Formation, including odobenids and phocids (*Ontocetus emmonsi*, *Phocanella pumila*, *Homiphoca* sp., *Platyphoca vulgaris*, *Callophoca obscura*, *Auroraphoca atlantica* and *Gryphoca similis*). Remarkably, in Pisco Formation, the seventh most prolific unit with nine fossil occurrences of extinct taxa, four species of phocids (*Acrophoca longirostris*, *Piscophoca pacifica*, *Hadrokirus martini* and *Australophoca changorum*) and one species of otariid (*Hydrarctos lomasiensis*) have been described.

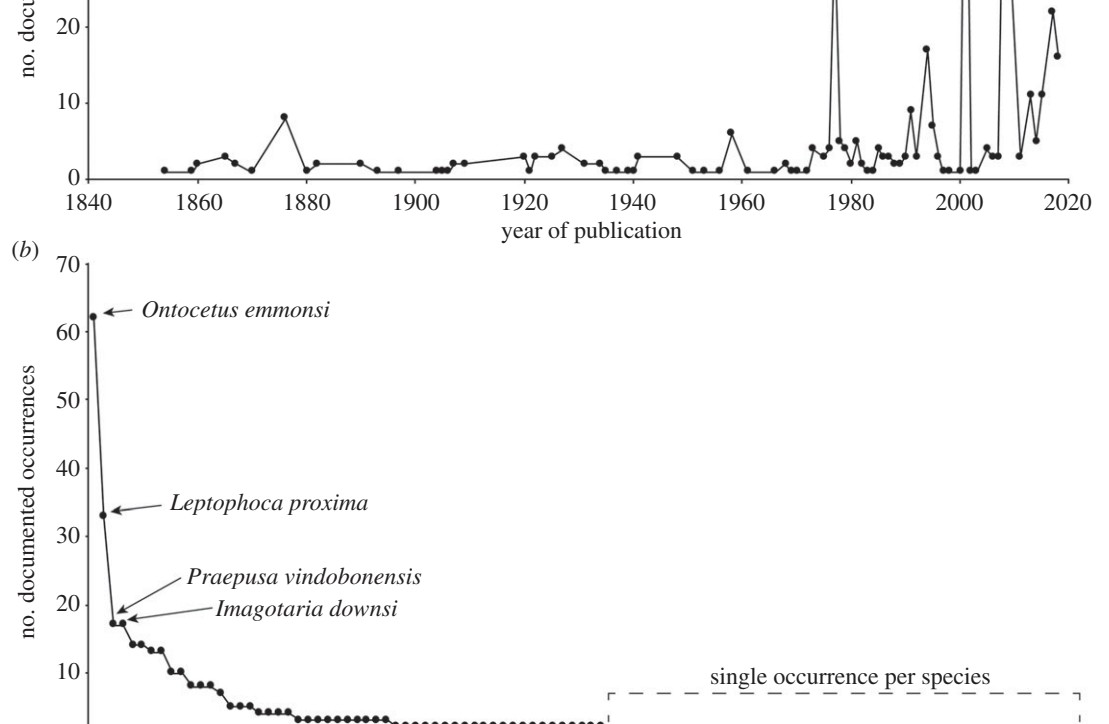

**Figure 4.** (*Caption overleaf.*)

Of the 388 occurrences of extinct pinnipeds identified at level of species, 50% ($n = 194$) of them belong to Phocidae, 30% ($n = 116$) belong to Odobenidae, 10% ($n = 37$) belong to Otariidae and Pan-Otariidae, 5% ($n = 20$) belong to Desmatophocidae and 5% ($n = 20$) belong to the so-called enaliarctids. Similarly, in terms of the taxonomic counts throughout the Cenozoic, of the 102 recognized extinct species, Phocidae constitute the largest taxonomically diverse clade, comprising 45 extinct species. Odobenidae is the second richest clade including 21 extinct species, contrasting with their monotypic extant diversity represented only by *O. rosmarus*. Otariidae and Pan-Otariidae comprise together 13 extinct species. The extinct clade Desmatophocidae comprises 11 species, and the so-called enaliarctids are represented by 10 species. Only *Palaeotaria henriettae* from the late Oligocene of France [44] was not assigned to any known group because the lack of unequivocal information to confirm its identity [45], remaining Pinnipedia *incertae sedis* in this study.

Overall, the odobenid *Ontocetus emmonsi* is the most represented extinct species constituting 16% ($n = 62$; figure 4b) of the total of the published fossil record of extinct species. Phocids, such as *L. proxima* and *Praepusa vindobonensis*, the odobenid *Imagotaria downsi* and the phocid *Homiphoca capensis*, are frequent findings in the fossil record, constituting 9% ($n = 33$), 4% ($n = 17$), 4% ($n = 17$) and 4% ($n = 14$), respectively (figure 4b). Notably, nearly half ($n = 49$) of the total of extinct species are established by a single fossil occurrence, which means the presence of a single specimen representing a single individual (figure 4b).

It should be noted that these results might point to a hidden diversity of other extinct species that are relatively abundant in the fossil record. For example, the phocid *Homiphoca capensis* from the Pliocene of

**Figure 4.** Fossil record of extinct species ($n = 388$). (*a*) Number of pinniped occurrences of extinct species through calendar years. We identified years with outstanding publications of fossil occurrences, which are indicated by numbers. 1: Repenning and Tedford (1977); 2: Koretsky (2001); 3: Kohno and Ray (2008). (*b*) Number of published occurrences per extinct species. Abbreviations: 1. *Ontocetus emmonsi* ($n = 62$); 2. *Leptophoca proxima* ($n = 33$); 3. *Praepusa vindobonensis* ($n = 17$); 4. *Imagotaria downsi* ($n = 17$); 5. *Cryptophoca maeotica* ($n = 14$); 6. *Homiphoca capensis* ($n = 14$); 7. *Callophoca obscura* ($n = 13$); 8. *Monachopsis pontica* ($n = 13$); 9. *Callorhinus gilmorei* ($n = 10$); 10. *Phocanella pumila* ($n = 10$); 11. *Dusignathus santacruzensis* ($n = 8$); 12. *Pliophoca etrusca* ($n = 8$); 13. *Pithanotaria starri* ($n = 8$); 14. *Pontophoca sarmatica* ($n = 7$); 15. *Gryphoca similis* ($n = 5$); 16. *Prophoca rousseaui* ($n = 5$); 17. *Allodesmus kernensis* ($n = 5$); 18. *Pachyphoca ukrainica* ($n = 4$); 19. *Thalassoleon mexicanus* ($n = 4$); 20. *Dusignathus seftoni* ($n = 4$); 21. *Thalassoleon macnallyae* ($n = 4$); 22. *Pteronarctos goedertae* ($n = 3$); 23. *Australophoca changorum* ($n = 3$); 24. *Praepusa pannonica* ($n = 3$); 25. *Platyphoca vulgaris* ($n = 3$); 26. *Batavipusa neerlandica* ($n = 3$); 27. *Desmatophoca oregonensis* ($n = 3$); 28. *Enaliarctos emlongi* ($n = 3$); 29. *Acrophoca longirostris* ($n = 3$); 30. *Allodesmus sinanoensis* ($n = 3$); 31. *Nanophoca vitulinoides* ($n = 3$); 32. *Enaliarctos mealsi* ($n = 2$); 33. *Piscophoca pacifica* ($n = 2$); 34. *Pteronarctos piersoni* ($n = 2$); 35. *Frissiphoca aberrantum* ($n = 2$); 36. *Pontolis magnum* ($n = 2$); 37. *Gomphotaria pugnax* ($n = 2$); 38. *Prototaria primigenia* ($n = 2$); 39. *Gryphoca nordica*; 40. *Titanotaria orangensis* ($n = 2$); 41. *Allodesmus demerei*; 42. *Pliopedia pacifica* ($n = 2$); 43. *Hydrarctos lomasiensis* ($n = 2$); 44. *Monotherium wymani* ($n = 2$); 45. *Proterozetes ulysses* ($n = 2$); 46. *Pelagiarctos thomasi* ($n = 2$); 47. *Enaliarctos mitchelli* ($n = 2$); 48. *Aivukus cedrosensis* ($n = 2$); 49. *Enaliarctos tedfordi* ($n = 2$); 50. *Pinnarctidion bishopi* ($n = 2$); 51. *Valenictus imperialensis* ($n = 2$); 52. *Pinnarctidion rayi* ($n = 2$); 53. *Phoca moori* ($n = 2$); 54. *Properyptychus argentinus* ($n = 1$); 55. *Brachyallodesmus packardi* ($n = 1$); 56. *Neotherium mirum* ($n = 1$); 57. *Virginiaphoca magurai* ($n = 1$); 58. *Noriphoca gaudini* ($n = 1$); 59. *Praepusa magyaricus* ($n = 1$); 60. *Odobenus mandanoensis* ($n = 1$); 61. *Prototaria planicephala* ($n = 1$); 62. *Allodesmus naorai* ($n = 1$); 63. *Eotaria citrica* ($n = 1$); 64. *Oriensarctos watasei* ($n = 1$); 65. *Pontophoca simionescui* ($n = 1$); 66. *Otaria fischeri* ($n = 1$); 67. *Atopotarus courseni* ($n = 1$); 68. *Pachyphoca chapskii* ($n = 1$); 69. *Miophoca vetusta* ($n = 1$); 70. *Desmatophoca brachycephala* ($n = 1$); 71. *Pseudotaria muramotoi* ($n = 1$); 72. *Pacificotaria hadromma* ($n = 1$); 73. *Thalassoleon inouei* ($n = 1$); 74. *Palaeotaria henriettae* ($n = 1$); 75. *Valenictus chulavistensis* ($n = 1$); 76. *Palmidophoca callirhoe* ($n = 1$); 77. *Archaeodobenus akamatsui* ($n = 1$); 78. *Eotaria crypta* ($n = 1$); 79. *Praepusa boeska* ($n = 1$); 80. *Hadrokirus martini* ($n = 1$); 81. *Enaliarctos barnesi* ($n = 81$); 82. *Histriophoca alekseevi* ($n = 1$); 83. *Proneotherium repenningi* ($n = 1$); 84. *Eumetopias kishidai* ($n = 1$); 85. *Messiphoca mauretanica* ($n = 1$); 86. *Frisiphoca affine* ($n = 1$); 87. *Protodobenus japonicus* ($n = 1$); 88. *Allodesmus kelloggi* ($n = 1$); 89. *Afrophoca libyca* ($n = 1$); 90. *Devinophoca claytoni* ($n = 1$); 91. *Auroraphoca atlantica* ($n = 1$); 92. *Platyphoca danica* ($n = 1$); 93. *Sarmatonectes sintsovi* ($n = 1$); 94. *Devinophoca emryi* ($n = 1$); 95. *Allodesmus megallos* ($n = 1$); 96. *Kamtschatarctos sinelnikovae* ($n = 1$); 97. *Necromites nestoris* ($n = 1$); 98. *Allodesmus uraiporensis* ($n = 1$); 99. *Neophoca palatina* ($n = 1$); 100. *Kawas benegasorum* ($n = 1$); 101. *Pontophoca jutlandica* ($n = 1$); 102. *Nanodobenus arandai* ($n = 1$).

South Africa is only represented by 14 fossil occurrences in electronic supplementary material, table S1. However, Govender *et al.* [46] identified 40 specimens belonging to this species and suggested that 'two or possible three seal taxa' existed among this material. Furthermore, Govender *et al.* [46] mentioned that more than 3000 specimens have been identified belonging to *H. capensis*, currently housed at the Iziko South African Museum, in Cape Town, South Africa, although a large portion remain undescribed [46]. A similar case is represented by the phocid *Callophoca obscura*, which is represented by an extensive fossil record from the western coast of North America (see electronic supplementary material, table S5); however, only 13 records of this species are included in electronic supplementary material, table S1. Considering this taxonomical uncertainty, we remain conservative and we only included the specimens recognized by the PaleoDB.

Finally, the most prolific author describing fossils of extinct taxa (including a variety of taxonomical ranks) is I. Koretsky, who is responsible for 20% of the taxonomic reporting. She is followed by N. Kohno (12%), C. A. Repenning (8%), L. Barnes (6%), A. Berta and R. Boessenecker (the latter two with 4%, each). To further explore in their efforts by geographical research area, we analysed the study sites for each of these authors and found that without exception the authors have described specimens from the Northern Hemisphere. In more detail, when analysed the geographical region of the specimens described for each of them, we noted a proclivity for those authors for working on fossil specimens from their own region versus other areas or continents (see electronic supplementary material, figure S4).

## 3.4. Type specimens of extinct species

It was not possible to state the catalogue number (i.e. the collection number of the research institution such as museums and universities in which the materials are deposited) for the type specimen (or holotype) of 13 extinct species: *Allodesmus sinanoensis*, *Eumetopias kishidai*, *Hydrarctos lomasiensis*, *Miophoca vetusta*, *Necromites nestoris*, *Oriensarctos watasei*, *Otaria fischeri*, *Pala. henriettae*, *Phoca (Phoca)*

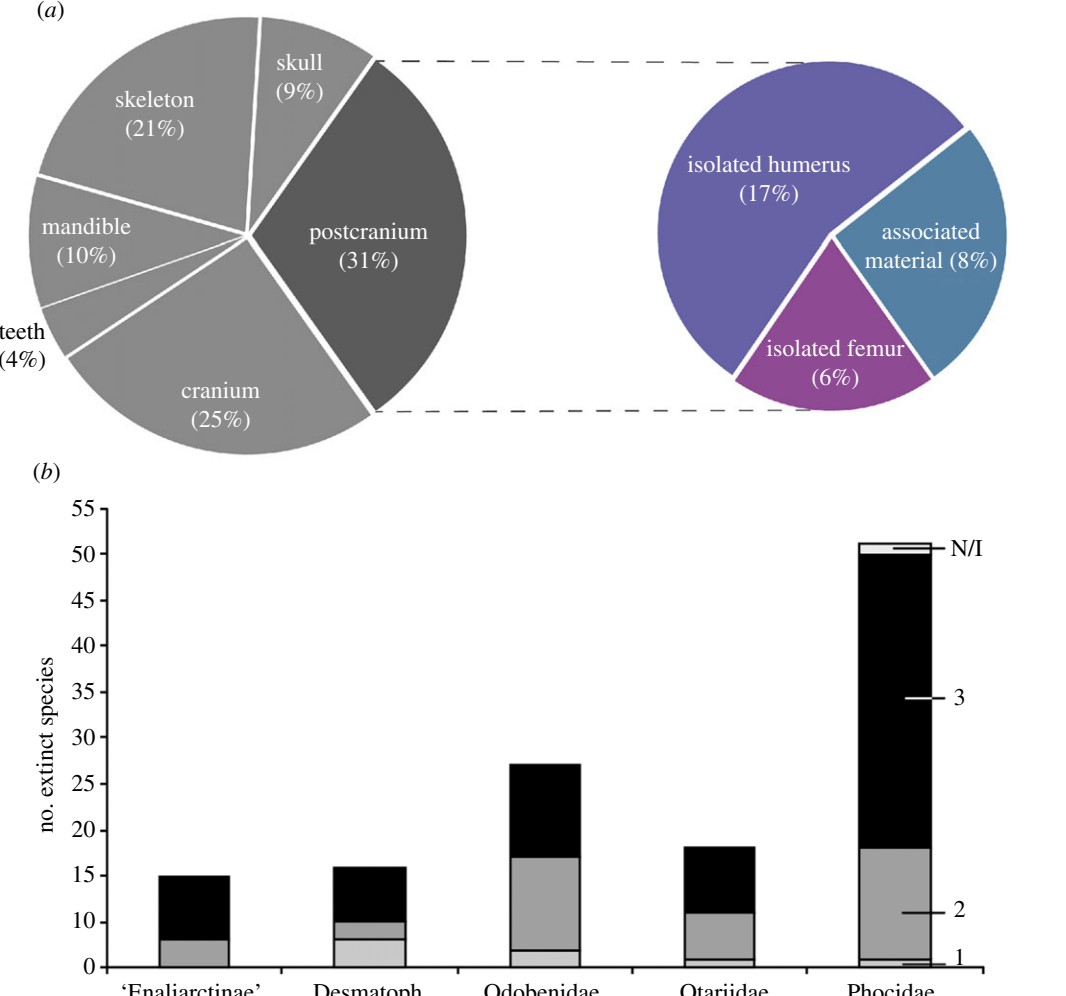

**Figure 5.** Features of the type material of extinct species ($n = 103$) recognized by the PaleoDB. (*a*) Identity of the type specimens of the extinct species. The bar representing postcranial remains shows the proportion of species which type materials is only represented by an isolated humerus (light grey), isolated femur (medium grey) and associated elements (dark grey). (*b*) Completeness of type specimens of extinct species by family. Light grey represents category 1 (complete or mostly complete skeletons), medium grey represents category 2 (associated elements) and dark grey represents category 3 (isolated elements). One species of phocid could not be assigned to any of the categories and it is stated as N/I.

*moori*, *Pontophoca sarmatica*, *Pontophoca simionescui*, *Praepusa vindobonensis* and *Protodobenus japonicus*. Some of them (*A. sinanoensis*, *E. kishidai*, *O. watasei* and *P. japonicus*) are housed at small institutions in Japan; therefore, our inability to state a catalogue number could be due to language constraints and limited accessibility to the specimens by researchers. In other cases, these taxonomic entities were erected in the nineteenth or early twentieth century without a designated type specimen, or remain problematic in other ways. Establishing their validity ought to be the focus of future systematic work.

Regarding the anatomical basis of these latter type specimens (figure 5*a*), 31% ($n = 32$) of extinct species have a holotype represented only by postcranial elements; 25% ($n = 26$) by cranium only, 21% ($n = 22$) by associated or articulated skeleton; 9% ($n = 9$) by skull (cranium and mandible); 10% ($n = 10$) by isolated mandibular remains; and 4% ($n = 4$) by dentition alone (figure 5*a*). We identified that the most common postcranial elements used as type specimen are isolated humeri and femora, representing a non-trivial quantity of 17% ($n = 18$) and 6% ($n = 6$), respectively (figure 5*a*). Similarly, 8% ($n = 8$) extinct species are based on type material constituted by associated postcranial elements. Thus, we identified that 23% of the extinct species of pinnipeds are founded only on isolated (and sometimes incomplete) appendicular elements as humeri and femora.

Regarding the completeness of type specimens of extinct species, we identified 61% ($n = 63$) belonging to category 3 (isolated elements), 31% ($n = 32$) to category 2 (associated or more complete remains), 7% ($n = 7$) to category 1 (complete or almost complete skeletons). For 1% ($n = 1$), it was not

possible to assign a category because of the missing information (i.e. *Mi. vetusta*). Therefore, the majority of fossil pinniped taxa are based on the type material that belong to a single isolated element (figure 5*b*).

# 4. Discussion

Uhen & Pyenson [10] were the first to examine the potential biases affecting diversity counts in fossil marine mammals, focusing on fossil cetaceans (with a brief comparison to fossil sirenians). They identified that taxonomic practices, such as the use of non-diagnostic-type specimens and taxonomic over-splitting, may artificially increase diversity indices. Other factors, such as the sedimentary outcrop area, sedimentation, preservation, collection and publication efforts, have the potential to bias diversity estimations [10]. Unfortunately for fossil pinnipeds, a few studies have directly addressed the influence of these factors, with the exception of Marx [11], who discussed, in part, the European fossil record of pinnipeds. While far from an exhaustive review of the potential biases affecting the pinniped fossil record and biodiversity estimations, our results show some relevant points.

## 4.1. Sampling biases of the pinniped fossil record

The effect of sampling biases (e.g. Pull of the Recent, outcrop area, among others) on diversity estimates from the fossil record has been the topic of extensive discussion in the paleobiological literature for nearly half a century (e.g. [47–52]; and references therein). In particular, it has been shown that the fossil record of cetaceans and sirenians lacks significant sampling bias, nor is there a display of strong the 'Pull of the Recent' (*sensu* [48]).

Our broad evaluation of the fossil occurrences of pinnipeds, along with their species richness through time, showed no correspondence with sedimentary outcrop area (figures 2 and 3; electronic supplementary material, table S3), suggesting that changes in the pinniped diversity may reflect genuine biological patterns rather than manifest geological factors [48,50,51]. Aside from this apparent lack of rock record bias, the pinniped record does appear to exhibit a Pull of the Recent trend with a relative increase in the number of species in younger time bins, towards the Recent (figure 2). This pattern is not surprising given broad-scale patterns in the marine tetrapod fossil record [18,53] that largely echo this trend in clades with larger numbers of extant and extinct species. Several authors have argued that environmental shifts increasing the patchiness and abundance of the primary productivity were likely strong factors in driving this trend in taxonomic and ecological diversity in both coastal and pelagic seas (see [54–56]). This argument seems to be supported by the timing of the diversification among otariids in the North Pacific during the late Miocene [57] and also the Mio-Pliocene record of odobenids [3].

Furthermore, we found that the most prolific geological units for fossil pinnipeds in the USA, Calvert and Yorktown formations did not show a correspondence with the number of species described in those units, nor with the number of fossils recovered from those collections (electronic supplementary material, tables S4 and S5). These results highlight that the amount of fossil remains recognized in geologic units and the number of species identified are not correlated, probably because the influence of taphonomic processes (among others) that limit the preservation of features with a diagnostic value (see below).

Taphonomy represents an important factor in the preservation of pinnipeds. Their amphibious lifestyle—equally dependent on land for breeding and on water for foraging—causes pinnipeds to be exposed to very different taphonomic processes depending on where they die (rookeries versus coastal or even pelagic environments). In this regard, we conceive three possible scenarios in which pinniped remains might enter into the fossil record: (i) death at sea and deposition offshore; (ii) death at sea followed by beaching of the carcass on the coast; and (iii) death and burial on land, either at haul out or rookery sites, or after stranding due to illness or injury. The potential effect of the taphonomic processes associated with these scenarios on the pinniped fossil record is largely unexplored, although they likely share some similarities with processes for other marine mammals, such as cetaceans (see [58]). The pre- and post-mortem processes that might accompany each of these modes of death are very different and are likely to affect the completeness of skeletons found, bone weathering state, bone modifications, the species composition at the site and the demography of sites with multiple individuals of the same species. While many studies have addressed taphonomic processes involving pinniped remains in part [6,19,59–63], future work needs to address and evaluate the full range of biotic and abiotic factors, as well intrinsic factors (e.g. body size, morphology, physiology, ontogeny and natural history) that potentially bias preservation across the entire clade of pinnipeds.

## 4.2. Geographical biases of the fossil record

Our analysis revealed that 89% of the published fossil record of pinnipeds (from the Cenozoic and Quaternary) is represented by occurrences from the Northern Hemisphere, denoting an enormous disparity in the location of the fossil findings between both hemispheres. There are several potential explanations for this dissimilarity, including differences in the past geographical distribution and abundance of the group (including the fact that pinnipeds originated along the North Pacific), as well other extrinsic explanations related with their fossilization and preservation; and other human-related factors including the sampling and collection effort and the likelihood of publication (e.g. [64–71]). Because of the unpredictability of the processes involved in the preservation and discovery of fossil remains, we cannot reject any of these potential explanations. Nevertheless, the stark differences in the published record from the eastern coasts of the North and South Pacific Ocean do deserve further consideration. Both regions exhibit analogous physical and ecological conditions, including high primary productivity [72,73] and the occurrence of a rich and diverse marine vertebrate fauna [74]. Notably, both regions exhibit broad fossiliferous sedimentary outcrops with a mostly remarkable contemporaneous fossil record of pinnipeds (and other marine mammals) from the middle Miocene and Pliocene (see [2] and references therein). Fossiliferous outcrops from the middle Miocene and Pliocene along the coast of California and along the western coast of South America both extend across a vast latitudinal gradient relatively constant range encompassing almost the entire southern coast of Peru and the northern and central coast of Chile, comprising two of the most important geologic units of the Southern Hemisphere: the Pisco and Bahía Inglesa formations in southern Peru and northern Chile, respectively [75,76]. In fact, both formations have yielded several species of marine vertebrates including, at least, four species of pinnipeds and still several species and morphotypes remain to be described (A.V.-T., personal observations), and rank among the world's richest fossil marine mammal faunas.

We propose that the disparity in the number of published fossil record of pinnipeds during the middle Miocene and Pliocene between the two regions is likely associated with differential sampling and publication efforts, along with the number of vertebrate paleontologists working in the field. For example, we detected that, the six most productive systematists were Northern Hemisphere-based researchers (electronic supplementary material, figure S2). Furthermore, these researchers show a strong tendency of publishing specimens from regions where they live and work. Thus, field explorations and publication efforts in the Southern Hemisphere may be a key factor explaining these differences. Overall, this suggests a very promising future for the study of fossil pinnipeds (and other marine mammals) in the Southern Hemisphere.

## 4.3. Fossil record of extant species

Surprisingly, the fossil record of extant species of pinnipeds represents 39% of the overall fossil record of this group (including Pleistocene and Holocene occurrences), contrasting with the proportions of the fossil record of other marine mammals, such as cetaceans and sirenians, in which the record of modern species represents a small fraction of the fossil record (e.g. only 2% of the fossil record of sirenians correspond to extant species; PaleoDB). Also, with a few exceptions, including remains of *O. rosmarus* from the Miocene of Belgium and *E. jubata* from the Pliocene of Japan [38,39], the fossil record of extant species is mostly referred to recent occurrences from the Holocene and Pleistocene (figure 1), exhibiting, in general, a very similar geographical distribution to contemporary patterns (a notable exception is constituted by a Ross seal (*Ommatophoca* rossii) from the early Pleistocene of New Zealand; [21]). In fact, in some regions, such as the eastern and western North American coast and Europe, the Pleistocene pinniped assemblage is identical in composition to the modern local fauna. By contrast, we found that fewer than 15 occurrences of extinct taxa occur in this interval. The relative paucity of Quaternary extinct species may explain the lack of knowledge about the evolutionary history of extant species.

Equally, we suggest that careful and exhaustive studies of pinniped faunas from the Pleistocene and Holocene might represent an exceptional opportunity to better understand the most historically recent community-wide changes in this clade, as exemplars for the response of marine mammals to changing environmental conditions over time (including shifting baselines from human hunting; see [6]). Furthermore, the use of biogeochemistry and molecular techniques, such as stable isotopes and ancient DNA, respectively, would provide new clues regarding how humans interacted and potentially affected different populations of marine mammals during the past (e.g. [31,77]), and has the potential to shed

light on questions as to what degree the modern distribution of pinnipeds is a reflection of their human-caused decimation in the past few hundred years [78,79].

## 4.4. Fossil record of extinct species

Our analysis highlights some features of the fossil record of extinct species that deserve further attention. First, nearly half of the extinct species are based on a single (published) occurrence confined to single time interval. While our study does not account for subsequent follow-up work that refers to more skeletal material or increases the geographical and stratigraphic range data of a novel taxon, this strong modal pattern to the description of fossil pinniped has a major impact on the historiography of paleobiological research with this group. In this regard, an important caveat is that these results do not necessarily represent the true plenitude of the fossil record of extinct species. In fact, we argue that there are discrepancies between the published fossil record and museum collections, which remain mostly unpublished (see the Results section for more details, and electronic supplementary material, table S5), much as 'dark data' obscure many patterns in the Cenozoic marine invertebrate record that co-occurs with fossil pinniped-bearing units of the west coast of North America [80].

Separately, 23% of the extinct species rely on type specimens consisting of a single isolated postcranial element (e.g. humeri or femora). We think this practice among systematists has largely not been grounded in a quantitative comparative context that accounts for the osteological variation and sexual dimorphism in some extant pinniped species [81–83]. To our knowledge, most of the studies addressing the osteological variation from sexual dimorphism in modern and fossil pinnipeds have been based on skulls (see [84–86]), with only a few studies testing such patterns using postcranial elements [8,83,87,88]. For instance, Koretsky [8] examined the humeri and femora of extinct and modern phocine seals, finding that most of the dimorphic variation between males and females is related to differences in the size (e.g. absolute length and length of the deltoid crest of the humerus), overall shape and proportions of the structures (e.g. dorsoventral thickness of the diaphysis and length and thickness of the neck of the femur) and the depth of muscle insertions (e.g. depth and shape of the coronoid fossa). Koretsky [8] did not find a correlation in the variation of those features among fossil and modern species of seals, showing a high specificity of the variation among seals. Based on the qualitative observations of the humeri of leopard seals (*Hydrurga leptonyx*) and a Weddell seal (*Leptonychotes weddellii*), Dewaele *et al.* [88] proposed that complete or nearly complete humeri can be considered as diagnostic bones to differentiate among monachinae phocids, although this was not tested statistically. More recently, Churchill & Uhen [83] performed an exhaustive morphometric analysis of the interspecific variation in the humeri and femora of phocids. They found the existence of large intraspecific variation, suggesting that the diagnostic efficacy of isolated femora and humeri is unjustified, questioning their validity as diagnostic elements for new species.

A more critical issue is the lack of a standardized approach for systematists to deal with non-associated elements referred to a new extinct taxon. Koretsky [8] asserted that a presumed 'principle of correlation of parts' and an 'ecomorphotype hypothesis' were sufficient to establish association, in the absence of taphonomic, stratigraphic or locality data. The 'ecomorphotype hypothesis' is based on the notion that the ecological niche of modern phocine seals is reflected in bones of the postcranial skeleton (humeri and femora) and mandible. It also assumes that extinct phocine seals have analogous morphological units to modern phocine species, making possible the recognition of fossil species from isolated postcranial elements and mandibles [8]. This author then defined five morphological groups of phocine seals, each comprising both modern and fossil species based on specific features of the mandible, humeri and femora. Unfortunately, no exhaustive ecomorphological study has been performed in modern pinnipeds, and the validity of this hypothesis remains to be tested, despite its weak paleoecological foundation. An example of the taxonomic consequences of this unsupported practice is represented by the recently described phocids *Terranectes magnus* and *T. parvus*, from the late Miocene age Eastover and St Marys formations of the eastern USA [89]. Despite the fragmentary nature of their holotypes (proximal half of the left humerus and a partial femur, respectively), Rahmat *et al.* [89] erected taxonomic names for these species based on non-overlapping and non-associated cranial, axial and appendicular elements (including some from different stratigraphic levels). Later, Dewaele *et al.* [88] considered both *T. magnus* and *T. parvus* to be a *nomen dubium*.

This practice (the use of non-associated elements) might lead not merely to artificial increases in taxonomic diversity, but also to inaccurate inferences about phylogenetic relationships among fossil pinnipeds. For instance, the extinct phocids *Prophoca proxima* and *Leptophoca lenis* are both species originally based on fragmentary isolated postcranial remains from the Miocene of Belgium. An exhaustive

re-description and phylogenetic analysis of *Pr. proxima* and *L. lenis* resulted in the synonymization of both species, with the proposition of a new combination *L. proxima* [37]. A similar case is represented by desmatophocids belonging to the genus *Allodesmus*, from the Round Mountain Silt of California. Three species of *Allodesmus* have been described (*Allodesmus gracilis*, *A. kelloggi* and *A. kernensis*; [90]), distinguished by the morphological variation on the mandibles (e.g. symphysial angle, interalveolar septa, depth of the masseteric fossa, shape of the coronoid process; [91,92]). However, it has been argued that these differences fall within levels of intraspecific variation in extant pinnipeds [93]. Thus, *A. kelloggi* and *A. gracilis* should be identified as junior synonyms of *A. kernensis* (following [90]). Further studies focusing on intraspecific morphological variation within extant and extinct species are needed to clarify these issues. These taxonomic re-evaluations (e.g. declaration of *nomina dubia* and junior synonyms) should add an additional complexity to the fossil record of this group since taxa, based on isolated and non-overlapping materials, are more prone to changes in their systematic and taxonomic identity. Similarly, taxa described longer ago are more prone to re-evaluations over time.

Finally, we could not pinpoint a specimen voucher for the holotype of 13 extinct species; in some cases, no type specimen had been formally designated, which is an occasional aspect of ninteenth-century taxonomic practices (see [10]). This situation is particularly relevant to the issues identified in our study because it creates a number of negative consequences, including lack of traceability with the identification, validation and comparability with other putative fossil occurrences (see [94]). It is incumbent on the community of practising systematic researchers to develop best practices in specimen-based research that prevent these problems (e.g. deposit fossil specimens in accessible natural history collections such as museums or research institutions; avoid studying specimens derived from illegal or informal trade, as well as those collected outside of established natural history museums). Surprisingly, many of these problems still occur among marine mammal researchers in the twenty-first century (see [95,96,97]).

# 5. Conclusion

We performed a historiographical revision of the published fossil record of pinnipeds based in the Paleobiology Database. Our historiographical analysis revealed several clear trends in the pinniped fossil record:

1. Most of the record is from deposits in the Northern Hemisphere, despite favourable paleoecological and depositional conditions in the Southern Hemisphere. There appears to be a strong collection and publication bias given strong imprint from the legacy of study in major centres of learning in the Northern Hemisphere, especially in North America and Europe, which have larger population centres, more researchers and longer traditions of study.
2. Nearly half of all fossil species are represented by a single occurrence—and a single specimen of a single individual. Additionally, a substantial number of extinct species have type material represented only by isolated postcranial elements, which might lead to artificial increases in taxonomic diversity. Considering these results, studies focusing on intra- and interspecific variation in the morphology of cranial and postcranial elements in modern and fossil taxa are needed.
3. Overall, despite the large geographical bias exhibited by the fossil record of pinnipeds, it is relatively adequately represented for evolutionary and paleoecological studies depending on the geographical region (and time interval) investigated.

Although this investigation is far from being exhaustive, it still provides a starting point for future research addressing changes in diversity, paleobiogeography and paleoecology in pinnipeds.

Data accessibility. Additional data are in the electronic supplementary material.

Authors' contributions. A.V.-T. and N.D.P. conceived and designed the research, analysed the data and wrote the paper.

Competing interests. We declare we have no competing interests.

Funding. This study was supported by CONICYT PCHA/Becas Chile, Doctoral Fellowship (grant no. 2016-72170286) to A.V.-T. and a Smithsonian Institution Graduate Fellowship. Additional funding for this work comes from a National Museum of Natural History (NMNH) Small Grant Award, and the Smithsonian Institution's Remington Kellogg Fund.

Acknowledgements. We would like to thank P. Koch and D. Costa (UCSC), J. Parham (CSUF), J. Velez-Juarbe (LACM), D. Bohaska (NMNH), M. Gilmour (UCSC), D. Glynn (UCSC), C. Ravelo (UCSC), F. Parada (U. Chile) and C. Gutstein (U. Chile) for providing comments and suggestions during the performance of this study, C. Carter (NMNH) for her help in the literature search. Additionally, we would like to thank four anonymous reviewers for their helpful comments that significantly improved the manuscript. The paper is Paleobiology Database contribution number 350.

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
