## [Reviewer comments · Royal Society Open Science]

Review History

RSOS-190234.R0 (Original submission)

Review form: Reviewer 1

Is the manuscript scientifically sound in its present form?

No

Are the interpretations and conclusions justified by the results?

Yes

Is the language acceptable?

Yes

Is it clear how to access all supporting data?

Yes

Do you have any ethical concerns with this paper?

No

Have you any concerns about statistical analyses in this paper?

No

Recommendation?

Reject

Comments to the Author(s)

In my opinion, this is a review of the fossil record based on a database, not a research article.

Review form: Reviewer 2 (Robert W. Boessenecker)

Is the manuscript scientifically sound in its present form?

Yes

Are the interpretations and conclusions justified by the results?

Yes

Is the language acceptable?

Yes

Is it clear how to access all supporting data?

Yes

Do you have any ethical concerns with this paper?

No

Have you any concerns about statistical analyses in this paper?

No

Recommendation?

Major revision is needed (please make suggestions in comments)

Comments to the Author(s)

Major Comments

This study is a critical advance in the study of fossil pinnipeds, and a 'sequel' in a way to the seminal study by Uhen and Pyenson (2007). I applaud the authors' efforts, and think this is a nice dataset that can be explored in some 'fun' and enlightening ways - and I have suggested numerous additional avenues of investigation that I would like to see either fulfilled or concretely addressed.

I am curious about a few other questions/patterns that could be addressed from this dataset:

1) Aside from the two most prolific units; some sort of a scatterplot showing the relationship of the number of fossil occurrences X the # of taxa per unit might be interesting. This would be helpful for those of us wanting to learn more about some of the other units aside from the Calvert/Yorktown.

2) Is there some way to visualize geographic bias? E.g. some sort of a map or graph showing the proclivity for particular authors to work on fossils from their own continent v. material from other continents.

3) Uhen and Pyenson also investigated lineage duration/taxon longevity; such an approach here should be easily done.

4) I share similar concerns about the quality of pinniped holotypes; why not reproduce a version of Uhen and Pyenson's figure 4, but breaking down skull v. cranium v. skeleton v. postcrania.

5) I'm also curious about reevaluation of certain taxa and declaration of nomina dubia, junior synonyms, etc.; is there anyway to quantify the number of changes in taxonomic opinion per taxon (which should be countable in the PBDB) and track this perhaps A) by decade or B) by author?

And lastly 6) Uhen and Pyenson ultimately used the dataset to assemble a graph showing richness, origination, and extinction rates for Pinnipedia. Is there a reason this was not attempted here?

I would very much like to see a different version of supp figure 1 where the # of occurrences is plotted by stage rather than by epoch, and perhaps leave out the Holocene. This would hopefully reflect some finer resolution; for example, based on the Pacific record, I would predict a dip in the number of occurrences during the early Pleistocene.

Is it possible to investigate/plot (perhaps with a subset of the data) inferred depositional environment or rock type v. number of occurrences or time? E.g. sandstone, siltstone, mudstone/shale, carbonate, nonmarine, etc.

I am seriously concerned by the number of Holocene "fossil" occurrences (n=459; 1/3 of the entire dataset). Most of these appear to be modern sightings of live animals or modern skeletons rather than actual fossil (or, rather, subfossil), although some do appear to be zooarchaeological. The distribution and publishing record of modern occurrences of live animals has no bearing on the publication/research history of fossilized examples. This part of the dataset really needs to be thoroughly vetted. Matter of fact, this exact point was brought up by Mark Uhen at the 2018 SVP meeting: most of these Holocene occurrences are not fossil/subfossil/zooarchaeological occurrences, and he pointed out in the Q&A session that these records do not belong in the dataset. I concur, and a cursory glance at article titles in the dataset for these records suggest they are studies of extant marine mammals and not fossil occurrences.

Minor Comments

Supplementary Figure 1 should absolutely be moved into the main text.

What do the authors mean by 'taxonomic validity'? This has a very specific meaning in taxonomy and refers only to whether or not a name is available, and confers zero information about the quality of a holotype specimen and therefore whether or not the taxon is even diagnosable (which is what we really care about). Please correct this.

The paper discusses extinct/extant species, but many extinct species are "neospecies" (in the sense that it is used in paleornithology) within extant genera. Perhaps there is some utility in investigating fossils that represent extant genera, but not extant species?

Regarding the “Ecomorphotype hypothesis” – I think somewhat more commentary is warranted, and add at least one sentence explaining the “logic” behind Koretsky’s principle (e.g. analogy with extant taxa; group I mandible must go with group I femur, etc.).

A cursory glance at the dataset (~10 minutes) found a number of errors:

- 1) A record of *Allodesmus* from the Oligocene Pysht Formation does not exist, and is not reported in Boessenecker and Churchill (2018). I suspect that this record stems from Hunt and Barnes (1994) and has been misattributed to our paper for some strange reason. If true, it was actually identified as “*Otariidae* indet.” (=Otarioidea or Pinnipedimorpha of later authors) and no relation to *Allodesmus* or *Desmatophocidae* was implied by these authors.
- 2) The maximum age of *Otocetus emmonsii* reported from South Carolina (Boessenecker et al., 2018) is far too old; the age range is about 1.8-1.1 Ma, which is squarely within the Calabrian stage; in the SOI it is shown with an incorrect maximum age of Piacenzian (3.6 Ma).
- 3) Paleobio database often lists the first, but incorrect, reference for a fossil assemblage. Boessenecker (2011) did not report any fossil pinnipeds; these were reported by Boessenecker (2013), a follow up to the 2011 study, which reported sharks/fish/birds etc.
- 4) *Titanotaria orangensis*, *Nanodobenus*, “*Desmatophocine A*” of Barnes (1972) and *Allodesmus* sp. cf. *sadoensis* (“*Desmatophocine B*” of Barnes, 1972) are all missing from the database.

I understand that these are probably errors made during entry into the PBDB, and therefore constitute an extra level of data corruption between publication and database entry. However, since the present study is about historiography and as of yet not a study of the efficacy of the PBDB, some vetting should probably be done to extinguish bogus fossil occurrences that have been entered improperly or attributed to the wrong publication.

Minor corrections

279-282: Many of these holotypes mentioned (4 out of 10) are housed at small Japanese institutions, which in fairness deserves commentary or mention.

324: delete “is”

350: represent (no ‘s’ on end)

359-360: Perhaps also include “bone modifications” here.

363: some additional helpful references here: Serran et al. 2008: “Massive death of pinnipeds 1200 years ago: taphonomic history of the “Lobos Site” ... *Quaternary International* 183:1:135-142

Pyenson et al. 2009 – taphonomy of Sharktooth hill Bonebed, *Geology*...

Bigelow, 1994: Occurrence of a squaloid shark....*Allodesmus*... *Journal of Paleontology* 68:3:680-684

Boessenecker and Perry, 2011: Mammalian bite marks on juvenile fur seal bones.... *Palaeos* 26:2:115-120

373: Probably very important to note that pinnipeds had a northern hemisphere origin, which will strongly influence the biogeographic distribution – this is overprinted by research and field effort bias.

378: correct to “Because of the”

395: What are the oldest remains in Peru and Chile? The Peruvian cetacean record goes back to the middle Miocene. On the PBDB most seem to be Serravallian/Langhian with one purported Aquitanian record from the Gaiman Formation from a conference abstract. This may not reflect publication bias against the southern hemisphere and might actually reflect a later arrival of phocids to the southern hemisphere. For example, pinnipeds of any sort are not present within the quite densely sampled record of marine vertebrates from the Oligocene and earliest Miocene Canterbury Basin sequence of NZ, and the earliest known pinnipeds from Australasia date to about the Miocene/Pliocene boundary in both Australia and NZ. If I recall correctly, South Africa shows a similar pattern.

400-404: Could you propose some discrete tests/ways to evaluate these hypotheses?

411-412: If this is the “Cape Kidnappers fur seal”, then this is a Holocene specimen that washed out of a Maori midden and is less than 700 years old. Citation: Weston, R. J., Repenning, C. A., and Fleming, C. A., 1973, Modern age of supposed Pliocene seal, *Arctocephalus caninus* Berry (= *Phocartos hookeri* Gray), from New Zealand: *New Zealand Jour. Sci.*, v. 16, no. 3, p. 591-598.

416-418: It is worth noting that in many places Pleistocene pinniped assemblages (especially in the North Pacific and Europe) are quite similar, if not identical, to the modern local fauna.

417: relative paucity

419-422: I think it would be fair to cite Boessenecker (2013: *Geodiversitas*) here as that paper includes such a discussion of Pleistocene pinniped faunas from the eastern North Pacific, as it regards Pliocene-Holocene faunal change.

458: perhaps change to “non-associated elements”

465: non-associated rather than associating

466: Rahmat et al. did not use the phrase ‘holotypic series’ and list most of the specimens as referred specimens, and this should be corrected – or perhaps correct to ‘hypodigm’. Further, Dewaele et al. (2018: *Royal Society Open Science*) considered *Terranectes* to be a nomen dubium; this absolutely should be explained here and cited.

467: Please provide an example of artificially increasing taxonomic diversity by the naming of non-comparable parts (e.g. *Leptophoca* is probably a good example). Furthermore, what about taxonomic oversplitting, such as is the case with *Allodesmus*, where one species has been given a new species name each time a *slightly* different complete skeleton has been unearthed? (e.g. *Allodesmus gracilis* + *kelloggi* = *A. kernensis*).

475: “widely acknowledged standards” – please provide a citation to a review paper on the topic; I believe several have just been published this year in *Geological Curator*.

Decision letter (RSOS-190234.R0)

24-Apr-2019

Dear Dr Valenzuela-Toro:

Manuscript ID RSOS-190234 entitled "What do we know about the fossil record of pinnipeds? A historiographic investigation" which you submitted to Royal Society Open Science, has been reviewed. The comments from reviewers are included at the bottom of this letter.

In view of the criticisms of the reviewers, the manuscript has been rejected in its current form. However, a new manuscript may be submitted which takes into consideration these comments.

Please note that resubmitting your manuscript does not guarantee eventual acceptance, and that your resubmission will be subject to peer review before a decision is made.

Your resubmitted manuscript should be submitted by 22-Oct-2019. If you are unable to submit by this date please contact the Editorial Office.

on behalf of Dr Julia Brenda Desojo (Associate Editor) and Kevin Padian (Subject Editor)
openscience@royalsociety.org

Editor Comments:

One review was a one-sentence recommendation to reject, but the grounds were not valid. Reviewer 2's review is extensive and thoughtful, and the authors should respond to all his points with equal care. In many ways the paper is as much a review as an analysis, but I think it is valuable because the data are thoughtfully considered in their various contexts, and the manuscript is very well written. The title as it stands implies "review," and if it could be changed to reflect the "analysis" part I suspect it would resonate better. I would recommend a "reject/resub" decision mainly because our "major revision" timeline is three weeks and you will want to take the time to run the zooarchaeological analyses separately, as Reviewer 2 suggests (I don't think the data should be eliminated, only analyzed separately).

One caveat, and please clarify: The "Pull of the Recent," as Raup defined it, does not have the meaning you use here (but I can't think of the correct term at the moment). As he defined it, the best example is Sphenodon: the clade is unknown in the fossil record since the Cretaceous, but

the presence of living Sphenodon "pulls" the range all through the Tertiary without a single fossil. I recognize that people sometimes misuse his original meaning, but even so ... and anyway might you actually have a true POTR example in the pinniped record?

Reviewers' Comments to Author:

Reviewer: 1

Comments to the Author(s)

In my opinion, this is a review of the fossil record based on a database, not a research article.

Reviewer: 2

Comments to the Author(s)

Major Comments

This study is a critical advance in the study of fossil pinnipeds, and a 'sequel' in a way to the seminal study by Uhen and Pyenson (2007). I applaud the authors' efforts, and think this is a nice dataset that can be explored in some 'fun' and enlightening ways - and I have suggested numerous additional avenues of investigation that I would like to see either fulfilled or concretely addressed.

I am curious about a few other questions/patterns that could be addressed from this dataset:

- 1) Aside from the two most prolific units; some sort of a scatterplot showing the relationship of the number of fossil occurrences X the # of taxa per unit might be interesting. This would be helpful for those of us wanting to learn more about some of the other units aside from the Calvert/Yorktown.
- 2) Is there some way to visualize geographic bias? E.g. some sort of a map or graph showing the proclivity for particular authors to work on fossils from their own continent v. material from other continents.
- 3) Uhen and Pyenson also investigated lineage duration/taxon longevity; such an approach here should be easily done.
- 4) I share similar concerns about the quality of pinniped holotypes; why not reproduce a version of Uhen and Pyenson's figure 4, but breaking down skull v. cranium v. skeleton v. postcrania.
- 5) I'm also curious about reevaluation of certain taxa and declaration of nomina dubia, junior synonyms, etc.; is there anyway to quantify the number of changes in taxonomic opinion per taxon (which should be countable in the PBDB) and track this perhaps A) by decade or B) by author?

And lastly 6) Uhen and Pyenson ultimately used the dataset to assemble a graph showing richness, origination, and extinction rates for Pinnipedia. Is there a reason this was not attempted here?

I would very much like to see a different version of supp figure 1 where the # of occurrences is plotted by stage rather than by epoch, and perhaps leave out the Holocene. This would hopefully reflect some finer resolution; for example, based on the Pacific record, I would predict a dip in the number of occurrences during the early Pleistocene.

Is it possible to investigate/plot (perhaps with a subset of the data) inferred depositional environment or rock type v. number of occurrences or time? E.g. sandstone, siltstone, mudstone/shale, carbonate, nonmarine, etc.

I am seriously concerned by the number of Holocene “fossil” occurrences (n=459; 1/3 of the entire dataset). Most of these appear to be modern sightings of live animals or modern skeletons rather than actual fossil (or, rather, subfossil), although some do appear to be zooarchaeological. The distribution and publishing record of modern occurrences of live animals has no bearing on the publication/research history of fossilized examples. This part of the dataset really needs to be thoroughly vetted. Matter of fact, this exact point was brought up by Mark Uhen at the 2018 SVP meeting: most of these Holocene occurrences are not fossil/subfossil/zooarchaeological occurrences, and he pointed out in the Q&A session that these records do not belong in the dataset. I concur, and a cursory glance at article titles in the dataset for these records suggest they are studies of extant marine mammals and not fossil occurrences.

Minor Comments

Supplementary Figure 1 should absolutely be moved into the main text.

What do the authors mean by ‘taxonomic validity’? This has a very specific meaning in taxonomy and refers only to whether or not a name is available, and confers zero information about the quality of a holotype specimen and therefore whether or not the taxon is even diagnosable (which is what we really care about). Please correct this.

The paper discusses extinct/extant species, but many extinct species are “neospecies” (in the sense that it is used in paleornithology) within extant genera. Perhaps there is some utility in investigating fossils that represent extant genera, but not extant species?

Regarding the “Ecomorphotype hypothesis” – I think somewhat more commentary is warranted, and add at least one sentence explaining the “logic” behind Koretsky’s principle (e.g. analogy with extant taxa; group I mandible must go with group I femur, etc.).

A cursory glance at the dataset (~10 minutes) found a number of errors:

- 1) A record of *Allodesmus* from the Oligocene Pysht Formation does not exist, and is not reported in Boessenecker and Churchill (2018). I suspect that this record stems from Hunt and Barnes (1994) and has been misattributed to our paper for some strange reason. If true, it was actually identified as “*Otariidae* indet.” (=Otarioidea or Pinnipedimorpha of later authors) and no relation to *Allodesmus* or *Desmatophocidae* was implied by these authors.
- 2) The maximum age of *Ontocetus emmonsii* reported from South Carolina (Boessenecker et al., 2018) is far too old; the age range is about 1.8-1.1 Ma, which is squarely within the Calabrian stage; in the SOI it is shown with an incorrect maximum age of Piacenzian (3.6 Ma).
- 3) Paleobio database often lists the first, but incorrect, reference for a fossil assemblage. Boessenecker (2011) did not report any fossil pinnipeds; these were reported by Boessenecker (2013), a follow up to the 2011 study, which reported sharks/fish/birds etc.
- 4) *Titanotaria orangensis*, *Nanodobenus*, “*Desmatophocine A*” of Barnes (1972) and *Allodesmus* sp. cf. *sadoensis* (“*Desmatophocine B*” of Barnes, 1972) are all missing from the database.

I understand that these are probably errors made during entry into the PBDB, and therefore

constitute an extra level of data corruption between publication and database entry. However, since the present study is about historiography and as of yet not a study of the efficacy of the PBDB, some vetting should probably be done to extinguish bogus fossil occurrences that have been entered improperly or attributed to the wrong publication.

Minor corrections

279-282: Many of these holotypes mentioned (4 out of 10) are housed at small Japanese institutions, which in fairness deserves commentary or mention.

324: delete "is"

350: represent (no 's' on end)

359-360: Perhaps also include "bone modifications" here.

363: some additional helpful references here: Serran et al. 2008: "Massive death of pinnipeds 1200 years ago: taphonomic history of the "Lobos Site"... Quaternary International 183:1:135-142

Pyenson et al. 2009 - taphonomy of Sharktooth hill Bonebed, Geology...

Bigelow, 1994: Occurrence of a squaloid shark....Allodesmus... Journal of Paleontology 68:3:680-684

Boessenecker and Perry, 2011: Mammalian bite marks on juvenile fur seal bones.... Palaios 26:2:115-120

373: Probably very important to note that pinnipeds had a northern hemisphere origin, which will strongly influence the biogeographic distribution - this is overprinted by research and field effort bias.

378: correct to "Because of the"

395: What are the oldest remains in Peru and Chile? The Peruvian cetacean record goes back to the middle Miocene. On the PBDB most seem to be Serravallian/Langhian with one purported Aquitanian record from the Gaiman Formation from a conference abstract. This may not reflect publication bias against the southern hemisphere and might actually reflect a later arrival of phocids to the southern hemisphere. For example, pinnipeds of any sort are not present within the quite densely sampled record of marine vertebrates from the Oligocene and earliest Miocene Canterbury Basin sequence of NZ, and the earliest known pinnipeds from Australasia date to about the Miocene/Pliocene boundary in both Australia and NZ. If I recall correctly, South Africa shows a similar pattern.

400-404: Could you propose some discrete tests/ways to evaluate these hypotheses?

411-412: If this is the "Cape Kidnappers fur seal", then this is a Holocene specimen that washed out of a Maori midden and is less than 700 years old. Citation: Weston, R. J., Repenning, C. A., and Fleming, C. A., 1973, Modern age of supposed Pliocene seal, *Arctocephalus caninus* Berry (=Phocartos hookeri Gray), from New Zealand: New Zealand Jour. Sci., v. 16, no. 3, p. 591-598.

416-418: It is worth noting that in many places Pleistocene pinniped assemblages (especially in the North Pacific and Europe) are quite similar, if not identical, to the modern local fauna.

417: relative paucity

419-422: I think it would be fair to cite Boessenecker (2013: *Geodiversitas*) here as that paper includes such a discussion of Pleistocene pinniped faunas from the eastern North Pacific, as it regards Pliocene-Holocene faunal change.

458: perhaps change to “non-associated elements”

465: non-associated rather than associating

466: Rahmat et al. did not use the phrase ‘holotypic series’ and list most of the specimens as referred specimens, and this should be corrected – or perhaps correct to ‘hypodigm’. Further, Dewaele et al. (2018: *Royal Society Open Science*) considered *Terranectes* to be a nomen dubium; this absolutely should be explained here and cited.

467: Please provide an example of artificially increasing taxonomic diversity by the naming of non-comparable parts (e.g. *Leptophoca* is probably a good example). Furthermore, what about taxonomic oversplitting, such as is the case with *Allodesmus*, where one species has been given a new species name each time a *slightly* different complete skeleton has been unearthed? (e.g. *Allodesmus gracilis* + *kelloggi* = *A. kernensis*).

475: “widely acknowledged standards” – please provide a citation to a review paper on the topic; I believe several have just been published this year in *Geological Curator*.

Author's Response to Decision Letter for (RSOS-190234.R0)

See Appendix A.

RSOS-191394.R0

Review form: Reviewer 3 (Annalisa Berta)

Is the manuscript scientifically sound in its present form?

Yes

Are the interpretations and conclusions justified by the results?

Yes

Is the language acceptable?

Yes

Do you have any ethical concerns with this paper?

No

Have you any concerns about statistical analyses in this paper?

Yes

Recommendation?

Accept with minor revision (please list in comments)

Comments to the Author(s)

Sent to both Editor and Author (Appendix B).

Review form: Reviewer 4

Is the manuscript scientifically sound in its present form?

No

Are the interpretations and conclusions justified by the results?

No

Is the language acceptable?

No

Do you have any ethical concerns with this paper?

No

Have you any concerns about statistical analyses in this paper?

No

Recommendation?

Major revision is needed (please make suggestions in comments)

Comments to the Author(s)

This manuscript represents a significant contribution to pinniped paleontology. The authors have provided quantitative data behind a statement very much thrown around by pinniped paleontologists: that the fossil record of pinnipeds is poor. The study does an excellent job outlining why this may be, with a highlight on future directions for the field.

The authors have done this by taking advantage of an excellent resource: the Paleobiology Database (PBD). However, due to the nature of PBD, some extra vetting of the raw data (Supplemental Table 1) needs to be done to improve the accuracy of the analysis (please refer to specifics in review (see Appendix C)). I also recommend the authors take a second look at their results and discussion before the next submission, to check whether the conclusions they are making from the data provided by the PBD make sense in relation to the literature. Most of these issues may be rectified by taking care around the use of the term "occurrences" that appears frequently in text to refer to data in Supp. Table 1. This is only in regards to minor statements in the text, and none of the larger results/conclusions should be affected.

There are a few major and minor corrections I have suggested, as well as corrections to the raw dataset.

Decision letter (RSOS-191394.R0)

28-Sep-2019

Dear Dr Valenzuela-Toro

On behalf of the Editor, I am pleased to inform you that your Manuscript RSOS-191394 entitled "What do we know about the fossil record of pinnipeds? A historiographic investigation" has been accepted for publication in Royal Society Open Science subject to minor revision in accordance with the referee suggestions. Please find the referees' comments at the end of this email.

The reviewers and Subject Editor have recommended publication, but also suggest some minor revisions to your manuscript. Therefore, I invite you to respond to the comments and revise your manuscript.

- Ethics statement

- Data accessibility

If you wish to submit your supporting data or code to Dryad (<http://datadryad.org/>), or modify your current submission to dryad, please use the following link:
<http://datadryad.org/submit?journalID=RSOS&manu=RSOS-191394>

- Competing interests

- Authors' contributions

- Acknowledgements

- Funding statement

Because the schedule for publication is very tight, it is a condition of publication that you submit the revised version of your manuscript before 07-Oct-2019. Please note that the revision deadline will expire at 00.00am on this date. If you do not think you will be able to meet this date please let me know immediately.

on behalf of Dr Julia Brenda Desajo (Associate Editor) and Kevin Padian (Subject Editor)
openscience@royalsociety.org

Reviewer comments to Author:
Reviewer: 3

Comments to the Author(s)
Sent to both Editor and Author

Reviewer: 4

Comments to the Author(s)

This manuscript represents a significant contribution to pinniped paleontology. The authors have provided quantitative data behind a statement very much thrown around by pinniped paleontologists: that the fossil record of pinnipeds is poor. The study does an excellent job outlining why this may be, with a highlight on future directions for the field.

The authors have done this by taking advantage of an excellent resource: the Paleobiology Database (PBD). However, due to the nature of PBD, some extra vetting of the raw data (Supplemental Table 1) needs to be done to improve the accuracy of the analysis (please refer to specifics in review). I also recommend the authors take a second look at their results and discussion before the next submission, to check whether the conclusions they are making from the data provided by the PBD make sense in relation to the literature. Most of these issues may be rectified by taking care around the use of the term "occurrences" that appears frequently in text to refer to data in Supp. Table 1. This is only in regards to minor statements in the text, and none of the larger results/conclusions should be affected.

There are a few major and minor corrections I have suggested, as well as corrections to the raw dataset.

Author's Response to Decision Letter for (RSOS-191394.R0)

See Appendix D.

Decision letter (RSOS-191394.R1)

22-Oct-2019

Dear Dr Valenzuela-Toro,

I am pleased to inform you that your manuscript entitled "What do we know about the fossil record of pinnipeds? A historiographic investigation" is now accepted for publication in Royal Society Open Science.

Kind regards,
Lianne Parkhouse
Editorial Coordinator
Royal Society Open Science
openscience@royalsociety.org

on behalf of Dr Julia Brenda Desojo (Associate Editor) and Kevin Padian (Subject Editor)
openscience@royalsociety.org

Appendix A

Dear Editors,

We resubmit the following manuscript, "What do we know about the fossil record of pinnipeds? A historiographic investigation" for consideration as an article in Royal Society Open Science.

We thank Reviewer 2 for comments that have dramatically improved the manuscript. This reviewer provided constructive and helpful comments. We followed the majority of the suggestions. Please see our reply to the reviewer below, where we have carefully addressed each comment.

We would like to suggest Reviewer 2 as a referee. Also, we propose the following list of additional referees, whose broad knowledge of marine mammal paleontology make them ideal reviewers for this manuscript. They are ranked in order of potential interest in the manuscript:

Jorge Velez-Juarbe (Los Angeles County Museum). Email: velezjuarbe@gmail.com

Erich Fitzgerald (Museum Victoria). Email: efitzgerald@museum.vic.gov.au

Robert Boessenecker (College of Charleston). Email: boesseneckerrw@cofc.edu

Leonard Dewaele (George Mason University). Email: ldewaele@gmu.edu

James Rule (Monash University). Email: james.rule@monash.edu

Sincerely,

Ana VALENZUELA-TORO*

Department of Ecology and Evolutionary Biology,

University of California Santa Cruz,

Coastal Biology Building,

130 McAllister Way, Santa Cruz, CA 95060 USA

*Corresponding author, E-mail: avalenzuela.toro@gmail.com

Nicholas D. PYENSON

Department of Paleobiology,

National Museum of Natural History,

Smithsonian Institution,

P.O. Box 37012, Washington DC, 20013 USA

Reviewers' Comments to Author:

Reviewer: 1

Comments to the Author(s)

In my opinion, this is a review of the fossil record based on a database, not a research article.

RESPONSE: No comments.

Reviewer: 2

Comments to the Author(s)

This study is a critical advance in the study of fossil pinnipeds, and a 'sequel' in a way to the seminal study by Uhen and Pyenson (2007). I applaud the authors' efforts, and think this is a nice dataset that can be explored in some 'fun' and enlightening ways - and I have suggested numerous additional avenues of investigation that I would like to see either fulfilled or concretely addressed.

I am curious about a few other questions/patterns that could be addressed from this dataset:

1) Aside from the two most prolific units; some sort of a scatterplot showing the relationship of the number of fossil occurrences X the # of taxa per unit might be interesting. This would be helpful for those of us wanting to learn more about some of the other units aside from the Calvert/Yorktown.

RESPONSE: We agree and we included accumulation (rarefaction) curves for Calvert, Astoria, Yorktown and Purisima formations (Supplementary Figure 3).

2) Is there some way to visualize geographic bias? E.g. some sort of a map or graph showing the proclivity for particular authors to work on fossils from their own continent v. material from other continents.

RESPONSE: Yes, we have included a new visual representation (Supplemental Figure 2) showing in more detail the geographic inclination in the study of fossil taxa by the six most prolific authors describing extinct pinnipeds. This new graph shows more clearly the tendency of those authors to study fossil remains from areas closer to their permanent residency versus others from other regions or continents.

3) Uhen and Pyenson also investigated lineage duration/taxon longevity; such an approach here should be easily done.

RESPONSE: We disagree on both counts: first, should not have conducted a study of lineage duration because this analysis is outside the scope of this paper. Our study attempts to provide a general analysis of the quality and modes of the fossil record of pinnipeds rather than an exhaustive analysis of the duration/taxon longevity, which is more of a question of evolutionary patterns in the fossil record. Second, we do not think it could easily be done, as such investigations require a robust, time-calibrated phylogeny, and detailed assessment of the stratigraphic range of the fossil taxa in question.

4) I share similar concerns about the quality of pinniped holotypes; why not reproduce a version of Uhen and Pyenson's figure 4, but breaking down skull v. cranium v. skeleton v. postcrania.

RESPONSE: A figure containing this information was included in the original submission (Figure 3A, B). However, we agree with the reviewer and changed the aesthetic of the figure to make it more clear (now Figure 5A). Additionally, we added a new figure (5B) addressing the completeness (articulated skeletons v. associated elements v. isolated elements) of the type specimens.

5) I'm also curious about reevaluation of certain taxa and declaration of nomina dubia, junior synonyms, etc.; is there any way to quantify the number of changes in taxonomic opinion per taxon (which should be countable in the PBDB) and track this perhaps A) by decade or B) by author?

RESPONSE: We agree with this suggestion, although we did not include a more in-depth analysis of this matter in our revision because we consider this issue to be outside the scope of this study. However, in our revision, we speculate that taxa described longer ago are more prone to reevaluations over time, and as a result, we expect that they are more exposed to changes in their systematic and taxonomic identity. We addressed this issue in the main text.

And lastly 6) Uhen and Pyenson ultimately used the dataset to assemble a graph showing richness, origination, and extinction rates for Pinnipedia. Is there a reason this was not attempted here?

RESPONSE: Again, as with the comment of lineage duration, evolutionary rates is a topic outside of the scope of the manuscript.

I would very much like to see a different version of supp figure 1 where the # of occurrences is plotted by stage rather than by epoch, and perhaps leave out the Holocene. This would hopefully reflect some finer resolution; for example, based on the Pacific record, I would predict a dip in the number of occurrences during the early Pleistocene.

RESPONSE: Agree. We modified Supplemental Figure 1 with the aim to provide a finer resolution of the geologic time, but also a more complete representation of the record, including extant and extinct species. Part A of this figure shows the number of occurrences of extinct and extant species over geologic time (by stages) of both extant and extinct species of pinnipeds.

Is it possible to investigate/plot (perhaps with a subset of the data) inferred depositional environment or rock type v. number of occurrences or time? E.g. sandstone, siltstone, mudstone/shale, carbonate, nonmarine, etc.

RESPONSE: We consider that this is outside the scope of this study.

I am seriously concerned by the number of Holocene "fossil" occurrences (n=459; 1/3 of the entire dataset). Most of these appear to be modern sightings of live animals or modern skeletons rather than actual fossil (or, rather, subfossil), although some do appear to be

zooarchaeological. The distribution and publishing record of modern occurrences of live animals has no bearing on the publication/research history of fossilized examples. This part of the dataset really needs to be thoroughly vetted. Matter of fact, this exact point was brought up by Mark Uhen at the 2018 SVP meeting: most of these Holocene occurrences are not fossil/subfossil/zooarchaeological occurrences, and he pointed out in the Q&A session that these records do not belong in the dataset. I concur, and a cursory glance at article titles in the dataset for these records suggest they are studies of extant marine mammals and not fossil occurrences.

RESPONSE: We agree with this concern. We updated the dataset used in the analysis and we manually removed false fossil and subfossil pinniped occurrences from the dataset. After this revision, the record of fossil (and subfossil) occurrences from the Pleistocene and Holocene includes 495 records (compared with 459 only from the Holocene from the last version). Some of the results obtained after this correction changed (with regard to the original version of this manuscript); however, the major trends were maintained. We strongly think that it is not necessary to ban these occurrences after this amendment because the Quaternary record of pinnipeds is an important component of the record that constitutes one of the few available evidence for understanding several questions regarding the origin, biogeography, and conservation of living species of pinnipeds.

Minor Comments

Supplementary Figure 1 should absolutely be moved into the main text.

RESPONSE: Agree. We moved Supplementary Figure 1 into the main text.

What do the authors mean by ‘taxonomic validity’? This has a very specific meaning in taxonomy and refers only to whether or not a name is available, and confers zero information about the quality of a holotype specimen and therefore whether or not the taxon is even diagnosable (which is what we really care about). Please correct this.

RESPONSE: Corrected.

The paper discusses extinct/extant species, but many extinct species are “neospecies” (in the sense that it is used in paleornithology) within extant genera. Perhaps there is some utility in investigating fossils that represent extant genera, but not extant species?

RESPONSE: We discussed only five extinct species belong to extant genera (*Callorhinus gilmorei*, *Histiophoca alekseevi*, *Neophoca palatina*, *Otaria fisheri*, and *Phoca moori*), representing ~5% of the total of extinct species. Furthermore, most of those species are also represented by a single occurrence (the only exception is *C. gilmorei* with 9 occurrences of referred material). Based on the low number of extinct species, we consider unnecessary to perform a different analysis based on the fossils that represent extant genera versus extant species. Nevertheless, we added a sentence in the main text about this issue.

Regarding the “Ecomorphotype hypothesis” – I think somewhat more commentary is warranted, and add at least one sentence explaining the “logic” behind Koretsky’s principle (e.g. analogy with extant taxa; group I mandible must go with group I femur, etc.).

RESPONSE: Agree. We added additional sentences explaining in more detail the fundamentals of the Ecomorphotype hypothesis.

A cursory glance at the dataset (~10 minutes) found a number of errors:

1) A record of *Allodesmus* from the Oligocene Pysht Formation does not exist, and is not reported in Boessenecker and Churchill (2018). I suspect that this record stems from Hunt and Barnes (1994) and has been misattributed to our paper for some strange reason. If true, it was actually identified as “Otariidae indet.” (=Otarioidea or Pinnipedimorpha of later authors) and no relation to *Allodesmus* or Desmatophocidae was implied by these authors.

RESPONSE: Corrected.

2) The maximum age of *Ontocetus emmonsii* reported from South Carolina (Boessenecker et al., 2018) is far too old; the age range is about 1.8-1.1 Ma, which is squarely within the Calabrian stage; in the SOI it is shown with an incorrect maximum age of Piacenzian (3.6 Ma).

RESPONSE: Corrected.

3) Paleobio database often lists the first, but incorrect, reference for a fossil assemblage. Boessenecker (2011) did not report any fossil pinnipeds; these were reported by Boessenecker (2013), a follow up to the 2011 study, which reported sharks/fish/birds etc.

RESPONSE: Corrected.

4) *Titanotaria orangensis*, *Nanodobenus*, “Desmatophocine A” of Barnes (1972) and *Allodesmus* sp. cf. *sadoensis* (“Desmatophocine B” of Barnes, 1972) are all missing from the database.

RESPONSE: *Titanotaria orangensis* was missing because this taxon was published in October 2018, 6 months later the data were downloaded from PBDB. The same holds for *Nanodobenus* (published in August 2018). We updated our analysis to April 29, 2019, so those taxa are now included.

I understand that these are probably errors made during entry into the PBDB, and therefore constitute an extra level of data corruption between publication and database entry. However, since the present study is about historiography and as of yet not a study of the efficacy of the PBDB, some vetting should probably be done to extinguish bogus fossil occurrences that have been entered improperly or attributed to the wrong publication.

RESPONSE: We agree with this statement; however, an evaluation of the data entry for the PBDB is outside the scope of this manuscript. Nevertheless, we performed an exhaustive revision (to the best of our abilities) of the data used in this study to reduce imprecisions in the data.

Minor corrections

279-282: Many of these holotypes mentioned (4 out of 10) are housed at small Japanese institutions, which in fairness deserves commentary or mention.

RESPONSE: Agree. We added a statement about this issue.

324: delete "is"

RESPONSE: Corrected.

350: represent (no 's' on end)

RESPONSE: Corrected.

359-360: Perhaps also include "bone modifications" here.

RESPONSE: Added.

363: some additional helpful references here: Serran et al. 2008: "Massive death of pinnipeds 1200 years ago: taphonomic history of the "Lobos Site"... Quaternary International 183:1:135-142
Pyenson et al. 2009 – taphonomy of Sharktooth hill Bonebed, Geology...
Bigelow, 1994: Occurrence of a squaloid shark...Allodesmus... Journal of Paleontology 68:3:680-684
Boessenecker and Perry, 2011: Mammalian bite marks on juvenile fur seal bones.... Palaios 26:2:115-120

RESPONSE: We have included these references in the new version of the manuscript.

373: Probably very important to note that pinnipeds had a northern hemisphere origin, which will strongly influence the biogeographic distribution – this is overprinted by research and field effort bias.

RESPONSE: Agree. We added a sentence including this point.

378: correct to "Because of the"

RESPONSE: Corrected.

395: What are the oldest remains in Peru and Chile? The Peruvian cetacean record goes back to the middle Miocene. On the PBDB most seem to be Serravallian/Langhian with one purported Aquitanian record from the Gaiman Formation from a conference abstract. This may not reflect publication bias against the southern hemisphere and might actually reflect a later arrival of phocids to the southern hemisphere. For example, pinnipeds of any sort are not present within the quite densely sampled record of marine vertebrates from the Oligocene and earliest Miocene Canterbury Basin sequence of NZ, and the earliest known pinnipeds from Australasia date to about the Miocene/Pliocene boundary in both Australia and NZ. If I recall correctly, South Africa shows a similar pattern.

RESPONSE: Agree. The pinniped fossil record from South America from the Neogene is referred mostly to occurrences from units from the middle Miocene to the Pliocene (more recent than the fossil record of pinnipeds from the eastern coast of the North Pacific, for instance). Thus, a general comparison of the pinniped fossil record from the Neogene from the coast of California and the western coast of South America is potentially biased by this temporal difference. We have added a sentence in the main text clarifying that we are only comparing contemporaneous fossil occurrences from the middle Miocene to the Pliocene from both regions.

400-404: Could you propose some discrete tests/ways to evaluate these hypotheses?

RESPONSE: At the moment, we see these hypotheses as good subjects for minor, future studies, beyond the scope of this paper.

411-412: If this is the "Cape Kidnappers fur seal", then this is a Holocene specimen that washed out of a Maori midden and is less than 700 years old. Citation: Weston, R. J., Repenning, C. A., and Fleming, C. A., 1973, Modern age of supposed Pliocene seal, *Arctocephalus caninus* Berry (= *Phocarcos hookeri* Gray), from New Zealand: *New Zealand Jour. Sci.*, v. 16, no. 3, p. 591-598.

RESPONSE: Corrected.

416-418: It is worth noting that in many places Pleistocene pinniped assemblages (especially in the North Pacific and Europe) are quite similar, if not identical, to the modern local fauna.

RESPONSE: Agree. We added a sentence with this observation.

417: relative paucity

RESPONSE: Corrected.

419-422: I think it would be fair to cite Boessenecker (2013: Geodiversitas) here as that paper includes such a discussion of Pleistocene pinniped faunas from the eastern North Pacific, as it regards Pliocene-Holocene faunal change.

RESPONSE: Agree, we added this reference there.

458: perhaps change to “non-associated elements”

RESPONSE: Corrected.

465: non-associated rather than associating

RESPONSE: Corrected.

466: Rahmat et al. did not use the phrase ‘holotypic series’ and list most of the specimens as referred specimens, and this should be corrected – or perhaps correct to ‘hypodigm’. Further, Dewaele et al. (2018: Royal Society Open Science) considered *Terranectes* to be a nomen dubium; this absolutely should be explained here and cited.

RESPONSE: Agree. We eliminated the phrase “holotypic series”. In addition, we addressed the new Taxonomic category of *Terranectes* from Dewaele et al. (2018).

467: Please provide an example of artificially increasing taxonomic diversity by the naming of non-comparable parts (e.g. *Leptophoca* is probably a good example). Furthermore, what about taxonomic oversplitting, such as is the case with *Allodesmus*, where one species has been given a new species name each time a *slightly* different complete skeleton has been unearthed? (e.g. *Allodesmus gracilis* + kelloggi = *A. kernensis*).

RESPONSE: We agree and have added an example of a case of artificially increased taxonomic diversity by the assignment of fragmentary elements. Also, we added a brief discussion about the potential taxonomic oversplit in *Allodesmus* from the Sharktooth Hill bonebed (Round Mountain Silt).

475: “widely acknowledged standards” – please provide a citation to a review paper on the topic; I believe several have just been published this year in *Geological Curator*.

RESPONSE: We have revised the text to mention best practices, as we think those words best fit the meaning implied by the suggestion. *Geological Curator*’s Group merely advocates guidelines for standards in UK collections, and we see little traction (or widespread knowledge) of these guidelines outside of this community. Instead, our manuscript focuses on examples of bad practices in specimen-based research, which may ameliorate some of the problems we have observed.

Appendix B

This ms. reviews the taxonomic, temporal and geographic record of pinnipeds and uses the PBDB to explore the record and its potential biases including diversity estimations. The ms is a nice complement to an earlier historiographic study of the cetacean and sirenian fossil record (Uhen and Pyenson, 2007). It is also a good critique of taxonomic problems that ensue when using non-associated elements and the untested “ecomorphological approach” as a basis for taxonomic identity of fossil pinnipeds (especially phocids) resulting in questionable (at best) phylogenetic relationships.

In this resubmission. the authors appear to have addressed reviewers’ concerns that have considerably improved the ms. I suggest only a few additional minor revisions.

Valenzuela and Koch 2019 is an abstract---given the ephemeral nature of abstracts it would be better to cite a paper.

Graphs employ different Geologic time scale (stages vs ages--epoch names)- for the benefit of the general reader and consistency I suggest using both stage as well as ages --epoch names for all.

Appendix C

This manuscript represents a significant contribution to pinniped paleontology. The authors have provided quantitative data behind a statement very much thrown around by pinniped paleontologists: that the fossil record of pinnipeds is poor. The study does an excellent job outlining why this may be, with a highlight on future directions for the field.

The authors have done this by taking advantage of an excellent resource: the Paleobiology Database (PBD). However, due to the nature of PBD, some extra vetting of the raw data (Supplemental Table 1) needs to be done to improve the accuracy of the analysis (please refer to specifics in review). I also recommend the authors take a second look at their results and discussion before the next submission, to check whether the conclusions they are making from the data provided by the PBD make sense in relation to the literature. Most of these issues may be rectified by taking care around the use of the term "occurrences" that appears frequently in text to refer to data in Supp. Table 1. This is only in regards to minor statements in the text, and none of the larger results/conclusions should be affected.

There are a few major and minor corrections I have suggested, as well as corrections to the raw dataset.

Major manuscript edits

- The use of the term "occurrence" seems to occasionally be conflated with the fossil record. The definition provided in the Methods section refers to geographical collections, but some of the time this is not what is being discussed in the text. Occasionally the use of "occurrences" in text is in reference to the literature, published specimens, or fossil sites. This has resulted in a lot of confusion while reviewing this manuscript, especially when several statements are made about the "occurrences" of specific species of fossil pinnipeds. Throughout the manuscript, it is occasionally unclear whether the actual published fossil record of pinnipeds is being discussed, or collection effort by museums (which is what the current definition could refer to). This issue is enhanced when assessing the raw data in Supplemental Table 1. For example: existing confusion about the age of particular sites in the literature means many listed "occurrences" are actually double-ups of records from the same site (just split into two time periods). This artificially inflates the number of "occurrences" being discussed. I advise the authors revise the use of "occurrences" in the text, to avoid this confusion. Alternatively, make it clear that occasionally, "occurrences" will be used as a proxy term for what is being discussed in text (e.g. the fossil record). If the authors go with the latter, there is at least one instance where this will be problematic (Page 13, the discussion of the most represented extinct species, see minor comments for this section).
- I would like to see the Methods section split up into the sub-heading format that the Results and Discussion sections are in, for clarity.
- Most critically, there are some major and minor problems with the occurrences listed in Supp. Table 1. These errors, while minor in most instances, affect the accuracy of the results. This is especially pertinent when it comes to conclusions drawn from occurrences from the Southern Hemisphere. I have made some specific suggestions for fixes below in the *Dataset corrections* section. But I recommend the authors take another look at the dataset beyond these suggestions to ensure the accuracy of the raw data.

Minor manuscript corrections

Page 3, Line 5-6: “The fossil record of pinnipeds is based on localities from the late Oligocene-to-early Miocene of the North Pacific Ocean...” This makes it sound like that pinnipeds only existed in the North Pacific during this time. However, *Afrophoca libyca* (Koretsky and Domning 2014) and *Noriphoca gaudini* (Dewaele *et al.* 2018) are known from the early Miocene of the Mediterranean/Paratethys. Please rephrase to account for this.

Page 8, Line 2: What about occurrences from the western coast of the South Atlantic Ocean (Argentina)? Specifically, *Properyptychus argentinus* (de Muizon and Bond 1982) and *Kawas benegasorum* (Cozzuol 2001). These should probably be included here, considering their importance as the earliest pinniped fossils in the Southern Hemisphere.

Page 12, Line 6-7: *Homiphoca capensis* should be *Homiphoca* sp.; see Dewaele *et al.* 2018 for discussion on Yorktown Formation records of *Homiphoca*.

Page 12, Line 6-7: This list is missing *Callophoca obscura*. Several occurrences of *C. obscura* from the Yorktown Formation are reported in Koretsky and Ray 2008. Recently, Dewaele *et al.* 2018 suggested that the humeri referred in Koretsky and Ray 2008 to *C. obscura* were the only ones reliably referred to the species. Despite this, there is still a substantial record of *C. obscura* from the Yorktown Formation, and hence it should be included here.

The following three comments are on the section of the manuscript discussing the most represented extinct pinnipeds in the fossil record (page 13), and are linked to the major comment on conflating “occurrence” data with the fossil record. I will use this section to outline why it could be problematic making inferences and conclusions on the actual fossil record using occurrence data.

Page 13, Lines 3-5: This list is in regards to the most represented extinct species in the fossil record. I find it odd that some of the most abundant fossil phocids (*Callophoca obscura* and *Homiphoca capensis*) aren't included in this list. There are certainly more published specimens of these species than the 17 for *Praepusa vindobonensis* and *Imagotaria downsi* (see Koretsky and Ray 2008, Dewaele *et al.* 2018, Avery and Klein 2011, Govender *et al.* 2011, Govender *et al.* 2012, Govender 2015, Govender 2015, Govender 2019, Hendey and Repenning 1972, de Muizon and Hendey 1980). I acknowledge that it is very likely due to the use of “occurrence” data from the PBD.

Page 13, Line 1-7: To elaborate the issue in the above point (and the major comments): the authors have used this section (and the accompanying Figure 4) to discuss the fossil record. However, the number of collections with fossils of these taxa (which is what “occurrences” currently refers to) are not representative of the published fossil record of these species. Due to the number of institutions in a smaller geographical area, European and American records may be inflated relative to other countries. This is problematic, and makes me confused about whether the authors are discussing the fossil record of pinnipeds, or collection efforts by institutions.

Page 13, Line 1-7: As a side note: taxonomic confusion (which is rife in the fossil pinniped literature) may also inflate or deflate the numbers of the most represented extinct species (due to bulk referral of material to a particular taxon on minimal basis, see Koretsky and Ray 2008 for an abundance of examples of this).

I highly recommend the authors revise the use of “occurrences” on this section (and the manuscript in general), and rephrase key sections to preface that occurrences in geographic collections are being discussed. Also, please add some discussion to the end of this section addressing the issues

raised above, and that the conclusions on the representation of particular fossil pinniped taxa based on this data should be taken with caution.

Page 14, Line 18: This is a long sentence; I would consider starting a new sentence at “and for 1%”.

Page 14, Line 20: This should be “consisting of” instead of “consistent in”.

Page 16, Line 8: This should be “units for fossil pinnipeds”, rather than “units in fossil pinnipeds”.

Page 16, Line 24: “Accompany” instead of “accompanying”.

Page 17, Line 6-7: The intrinsic factors listed here are not expanded on, despite taphonomy being discussed at length in the paragraph. Unless there is a reason it is outside the scope of the paper, I would like to see a bit more discussion on how these factors influence the preservation of pinniped fossils. This would be a critical discussion to have, as it has not been addressed in full in other recent papers.

Page 18, Line 10-11: Please state whom this pers. communication is with. It could be any number of current marine mammal palaeontologists working with these fossils.

Page 18, Line 17: This should be “where they live”, not “where their live”.

Page 18, Line 19: This should be “efforts in the Southern Hemisphere”, not “efforts on the Southern Hemisphere”.

Page 19, Line 1: “such” should be added at the beginning of this line (before “as cetaceans and sirenians”).

Page 19, Line 7: A comment on the early Pleistocene fossil mandible of *Ommatophoca rossii* from New Zealand (King 1973) may be worth noting here, as it is well outside their current tight circumpolar distribution.

Page 19, Line 7: Insert “such” between “regions as” (“regions such as”).

Page 19, Line 17: Insert “such” between “techniques as” (“techniques such as”).

Page 19, Line 21: “reflection” instead of “reflects”.

Page 20, Line 11: Loza *et al.* 2015 is also a useful reference to cite here.

Page 20, Line 22: The authors have incorrectly stated that Dewaele *et al.* 2018 looked at Crabeater Seals (*Lobodon carcinophagus*) when doing qualitative observations. It was actually a Weddell Seal humerus (*Leoptonychotes weddellii*).

Page 21, Line 2: The recently published paper by Churchill and Uhen (2019) is only briefly discussed here. This paper is a significant advance in the area of phocid taxonomy, and has hard data behind the problems with using isolated postcrania as type specimens. I would like to see a little more nuanced discussion on what they found in terms of the validity of humeri and femora as diagnostic elements.

Page 21, Line 15: “of the taxonomic consequences” instead of “of this taxonomic consequences”.

Page 21, Line 24: “taxonomic diversity, and also” instead of “taxonomic diversity but also”. Alternatively, remove “also might” on Line 23 and replace with “might not only”.

Page 22, Line 1: Remove the second occurrence of “are” (“species are originally”).

Page 22, Line 19-21: Please cite some examples at the end of this sentence.

Page 22, Line 24: “systematic” instead of “systematics”.

Page 23, Line 1-3: The example in the brackets needs to be rewritten; as it is wordy and has a few double ups of terminology. Could potentially rewrite as: “(e.g. deposit of fossil specimens in valid natural history collections or research institutions, avoidance of studying specimens derived from illegal or informal trade, as well as from amateur private collectors)”.

Page 23, Line 3: Additionally, I find the inclusion of “amateur private collectors” here problematic. In a lot of countries, collections are primarily built by donations from amateur private collectors. Please reword to clarify what you mean (perhaps poachers, or collectors who do not work in conjunction with natural history museums), or consider removing.

Dataset corrections

1. Occurrences of *Callophoca obscura* are completely missing for the Yorktown Formation from Supplementary Table 1, despite being present in Supplementary Table 5. In addition, it is missing from the list of Yorktown Formation pinnipeds on Page 12, Line 6-7.
2. A few dates need revising in Supp Table 1:
 - *Virginiaphoca magurai* from the Eastover Formation would more specifically originate from the Cobham Bay Member. This is dated to 8.7-6.46 Ma. (Ward and Blackwelder 1980. Stratigraphic revision of upper Miocene and lower Pliocene beds of the Chesapeake group, middle Atlantic Coastal Plain; Dewaele *et al.* 2018. *Diversity of late Neogene Monachinae (Carnivora, Phocidae) from the North Atlantic, with the description of two new species*).
 - Keep in mind for the *Properiptychus argentinus* records that the paper the PB database references (Cione *et al.* 2000) for the middle Miocene age (13.82-11.62) of the Parana Formation does not present a strong case for this age. It is more likely Tortonian in age; see Perez *et al.* 2010. *Paleoecological and paleobiogeographic significance of two new species of bivalves in the Parana Formation (late Miocene) of entre rios province argentina*
 - The minimum age for *Noriphoca gaudini* is incorrect, it should be 20.44 Ma, not 2.588 Ma. See Dewaele *et al.* 2018 *A critical revision of the fossil record, stratigraphy and diversity of the Neogene seal genus Monotherium (Carnivora, Phocidae)*.
 - Doublecheck the maximum and minimum age records for *Nanophoca vitulinoides*, as sediment samples in Dewaele *et al.* 2017 have been dated to 14.2-11.63 Ma.
 - All records for the genus *Frisiphoca* should be conservatively dated to 11.6-5.3 Ma, see Dewaele *et al.* 2018. (*A critical revision of the fossil record, stratigraphy and diversity of the Neogene seal genus Monotherium (Carnivora, Phocidae)*) for a more detailed discussion.
 - Specimens from the Yorktown Formation may need to be dated to 4.9-3.9 Ma; see discussion of dating in the Supplemental Information of Marx and Fordyce 2015 (*Baleen boom and bust: a synthesis of mysticete phylogeny, diversity and disparity*).
3. A few records from Oceania in Supp Table 1 are erroneous. These are critical to correct given the dearth of records from the Southern Hemisphere:
 - No early Miocene phocids were reported from the Port Campbell Limestone of Australia in Fitzgerald 2005.
 - Additionally, phocids from Beaumaris (Australia) reported in Fitzgerald 2004 are the same record as the Monachinae from Fordyce and Flannery 1983.
 - The phocids from New Zealand referred to in McKee 1988 are the same phocids from the

Tangahoe Formation in McKee 1994. Keep in mind though, the specimens referenced to in these publications (McKee 1988, McKee 1994) are in a private collection, and so no publically accessible specimens from the Tangahoe formation have been published.

- The Waipunga miroungin specimen from New Zealand (Boessenecker and Churchill 2016) should be dated to 2.4-2.1 Ma, not 3.6-2.588 Ma (the "Late Pliocene" age referred to in the title of this paper is in reference to an older geological timescale, making this an early Pleistocene occurrence).

4. Double check the records and occurrences for *Properiptychus argentinus* in Supp Table 1, as they may represent the same fossils. Additionally, de Muizon and Bond 1982 is in French, not English.
5. The new paper recently published (Velez-Juarbe and Valenzuela-Toro 2019) on two monachine fossils from the Monterey Formation (Tortonian age) in the North Pacific would be useful to include in the dataset. If the analyses need to be re-run, it will be worth including in Supp Table 1 (provided they have a record in the PB database).

Appendix D

Dear Editors,

We submit the final edits of the manuscript “What do we know about the fossil record of pinnipeds? A historiographic investigation” which was accepted with minor revisions as an article in Royal Society Open Science.

We thank Reviewer 3 and 4 for comments that have significantly improved the manuscript. We followed the majority of the suggestions. Please see our reply to the reviewer below, where we have carefully addressed each comment.

Sincerely,

Ana Valenzuela-Toro and Nick Pyenson

--

Reviewers' Comments to Author:

Reviewer 3:

This ms. reviews the taxonomic, temporal and geographic record of pinnipeds and uses the PBDB to explore the record and its potential biases including diversity estimations. The ms is a nice complement to an earlier historiographic study of the cetacean and sirenian fossil record (Uhen and Pyenson, 2007). It is also a good critique of taxonomic problems that ensue when using non-associated elements and the untested “ecomorphological approach” as a basis for taxonomic identity of fossil pinnipeds (especially phocids) resulting in questionable (at best) phylogenetic relationships.

In this resubmission. the authors appear to have addressed reviewers' concerns that have considerably improved the ms. I suggest only a few additional minor revisions. Valenzuela and Koch 2019 is an abstract---given the ephemeral nature of abstracts it would be better to cite a paper.

RESPONSE: We agree. We have removed this reference from the main document.

Graphs employ different Geologic time scale (stages vs ages--epoch names)- for the benefit of the general reader and consistency I suggest using both stage as well as ages --epoch names for all.

RESPONSE: We agree, and we modified the figures, including both, geologic stages and epoch names.

Reviewer 4:

This manuscript represents a significant contribution to pinniped paleontology. The authors have provided quantitative data behind a statement very much thrown around by pinniped paleontologists: that the fossil record of pinnipeds is poor. The study does an excellent job outlining why this may be, with a highlight on future directions for the field. The authors have done this by taking advantage of an excellent resource: the Paleobiology Database (PBD). However, due to the nature of PBD, some extra vetting of the raw data (Supplemental Table 1) needs to be done to improve the accuracy of the analysis (please refer to specifics in review). I also recommend the authors take a second look at their results and discussion before the next submission, to check whether the conclusions they are making from the data provided by the PBD make sense in relation to the literature. Most of these issues may be rectified by taking

care around the use of the term "occurrences" that appears frequently in text to refer to data in Supp. Table 1.

This is only in regards to minor statements in the text, and none of the larger results/conclusions should be affected. There are a few major and minor corrections I have suggested, as well as corrections to the raw dataset.

RESPONSE: Thanks a lot for the constructive comments. We have addressed each of the observations below.

Major manuscript edits

- The use of the term "occurrence" seems to occasionally be conflated with the fossil record. The definition provided in the Methods section refers to geographical collections, but some of the time this is not what is being discussed in the text. Occasionally the use of "occurrences" in text is in reference to the literature, published specimens, or fossil sites. This has resulted in a lot of confusion while reviewing this manuscript, especially when several statements are made about the "occurrences" of specific species of fossil pinnipeds. Throughout the manuscript, it is occasionally unclear whether the actual published fossil record of pinnipeds is being discussed, or collection effort by museums (which is what the current definition could refer to).

This issue is enhanced when assessing the raw data in Supplemental Table 1. For example: existing confusion about the age of particular sites in the literature means many listed "occurrences" are actually double-ups of records from the same site (just split into two time Periods). This artificially inflates the number of "occurrences" being discussed. I advise the authors revise the use of "occurrences" in the text, to avoid this confusion. Alternatively, make it clear that occasionally, "occurrences" will be used as a proxy term for what is being discussed in text (e.g. the fossil record). If the authors go with the latter, there is at least one instance where this will be problematic (Page 13, the discussion of the most represented extinct species, see minor comments for this section).

RESPONSE: We agree with this observation. We have clarified our definition of "occurrences." We also made some edits along the text to make it clear when using a different definition of "occurrences" than the description included in the Methodology section.

- I would like to see the Methods section split up into the sub-heading format that the Results and Discussion sections are in, for clarity.

RESPONSE: We agree and we have added sub-heading titles in the Methods section.

- Most critically, there are some major and minor problems with the occurrences listed in Supp. Table 1. These errors, while minor in most instances, affect the accuracy of the results. This is especially pertinent when it comes to conclusions drawn from occurrences from the Southern Hemisphere. I have made some specific suggestions for fixes below in the Dataset corrections section. But I recommend the authors take another look at the dataset beyond these suggestions to ensure the accuracy of the raw data.

RESPONSE: See our specific response to each point below.

Minor manuscript corrections:

Page 3, Line 5-6: "The fossil record of pinnipeds is based on localities from the late Oligocene-to-early Miocene of the North Pacific Ocean..." This makes it sound like that pinnipeds only

existed in the North Pacific during this time. However, *Afrophoca libyca* (Koretsky and Domning 2014) and *Noriphoca gaudini* (Dewaele et al. 2018) are known from the early Miocene of the Mediterranean/Paratethys. Please rephrase to account for this.

RESPONSE: Agree. We have rephrased this sentence to account for these occurrences.

Page 8, Line 2: What about occurrences from the western coast of the South Atlantic Ocean (Argentina)? Specifically, *Properyptychus argentinus* (de Muizon and Bond 1982) and *Kawas benegasorum* (Cozzuol 2001). These should probably be included here, considering their importance as the earliest pinniped fossils in the Southern Hemisphere.

RESPONSE: Agree. We added the reference of those species in this section.

Page 12, Line 6-7: *Homiphoca capensis* should be *Homiphoca* sp.; see Dewaele et al. 2018 for discussion on Yorktown Formation records of *Homiphoca*.

RESPONSE: Agree. We corrected the sentence.

Page 12, Line 6-7: This list is missing *Callophoca obscura*. Several occurrences of *C. obscura* from the Yorktown Formation are reported in Koretsky and Ray 2008. Recently, Dewaele et al. 2018 suggested that the humeri referred in Koretsky and Ray 2008 to *C. obscura* were the only ones reliably referred to the species. Despite this, there is still a substantial record of *C. obscura* from the Yorktown Formation, and hence it should be included here.

RESPONSE: Agree. We have added *Callophoca obscura* to the list provided in page 12 of the main text.

The following three comments are on the section of the manuscript discussing the most represented extinct pinnipeds in the fossil record (page 13), and are linked to the major comment on conflating “occurrence” data with the fossil record. I will use this section to outline why it could be problematic making inferences and conclusions on the actual fossil record using occurrence data.

Page 13, Lines 3-5: This list is in regards to the most represented extinct species in the fossil record. I find it odd that some of the most abundant fossil phocids (*Callophoca obscura* and *Homiphoca capensis*) aren’t included in this list. There are certainly more published specimens of these species than the 17 for *Praepusa vindobonensis* and *Imagotaria downsi* (see Koretsky and Ray 2008, Dewaele et al. 2018, Avery and Klein 2011, Govender et al. 2011, Govender et al. 2012, Govender 2015, Govender 2015, Govender 2019, Hendey and Repenning 1972, de Muizon and Hendey 1980). I acknowledge that it is very likely due to the use of “occurrence” data from the PBD.

RESPONSE: Partially agree. We recognize that there are several fossil occurrences of both *Callophoca obscura* and *Homiphoca capensis*; however, most of those fossil remains have not been formally described in peer-reviewed literature. In this regard, we prefer to keep a conservative approach and decided to not include them in the Supplemental Table 1. Nevertheless, we added some sentences in the main text (page 13-14) recognizing this issue and making the reading aware that the results presented are only referred to fossil occurrences reported in the literature.

Page 13, Line 1-7: To elaborate the issue in the above point (and the major comments): the authors have used this section (and the accompanying Figure 4) to discuss the fossil record. However, the number of collections with fossils of these taxa (which is what “occurrences” currently refers to) are not representative of the published fossil record of these species. Due to the number of institutions in a smaller geographical area, European and American records may be inflated relative to other countries. This is problematic, and makes me confused about

whether the authors are discussing the fossil record of pinnipeds, or collection efforts by institutions.

RESPONSE: Agree. We have modified the main text (page 13) with the aim to clarify this concern.

Page 13, Line 1-7: As a side note: taxonomic confusion (which is rife in the fossil pinniped literature) may also inflate or deflate the numbers of the most represented extinct species (due to bulk referral of material to a particular taxon on minimal basis, see Koretsky and Ray 2008 for an abundance of examples of this). I highly recommend the authors revise the use of “occurrences” on this section (and the manuscript in general), and rephrase key sections to preface that occurrences in geographic collections are being discussed. Also, please add some discussion to the end of this section addressing the issues raised above, and that the conclusions on the representation of particular fossil pinniped taxa based on this data should be taken with caution.

RESPONSE: Agree. We added some sentences clarifying this issue in the main text (page 21).

Page 14, Line 18: This is a long sentence; I would consider starting a new sentence at “and for 1%”.

RESPONSE: We agree and we have modified the sentence.

Page 14, Line 20: This should be “consisting of” instead of “consistent in”.

RESPONSE: We agree and we have modified the sentence.

Page 16, Line 8: This should be “units for fossil pinnipeds”, rather than “units in fossil pinnipeds”.

RESPONSE: We agree and we have modified the sentence.

Page 16, Line 24: “Accompany” instead of “accompanying”.

RESPONSE: We agree and we have modified the sentence.

Page 17, Line 6-7: The intrinsic factors listed here are not expanded on, despite taphonomy being discussed at length in the paragraph. Unless there is a reason it is outside the scope of the paper, I would like to see a bit more discussion on how these factors influence the preservation of pinniped fossils. This would be a critical discussion to have, as it has not been addressed in full in other recent papers.

RESPONSE: We agree that a discussion about these topics is very relevant to the field. However, we consider that the suggestion of adding a more thoughtful discussion about how intrinsic factors might affect the preservation of pinniped fossils it is outside the scope of this paper. Nevertheless, one of the authors of this study (A.V.T.) is currently performing field-based research that will shed light on this critical matter.

Page 18, Line 10-11: Please state whom this pers. communication is with. It could be any number of current marine mammal palaeontologists working with these fossils.

RESPONSE: We have modified the sentence and added more detail about this personal observation.

Page 18, Line 17: This should be “where they live”, not “where their live”.

RESPONSE: We agree and we have corrected the spelling.

Page 18, Line 19: This should be “efforts in the Southern Hemisphere”, not “efforts on the Southern Hemisphere”.

RESPONSE: We agree and we have corrected the spelling.

Page 19, Line 1: “such” should be added at the beginning of this line (before “as cetaceans and sirenians”).

RESPONSE: We agree and we have modified the sentence.

Page 19, Line 7: A comment on the early Pleistocene fossil mandible of *Ommatophoca rossii* from New Zealand (King 1973) may be worth noting here, as it is well outside their current tight circumpolar distribution.

RESPONSE: We agree, and we have added a sentence about this occurrence in the main text (page 20).

Page 19, Line 7: Insert “such” between “regions as” (“regions such as”).

RESPONSE: We agree and we have modified the sentence.

Page 19, Line 17: Insert “such” between “techniques as” (“techniques such as”).

RESPONSE: We have modified the sentence.

Page 19, Line 21: “reflection” instead of “reflects”.

RESPONSE: We have modified the sentence.

Page 20, Line 11: Loza et al. 2015 is also a useful reference to cite here.

RESPONSE: We have added the reference.

Page 20, Line 22: The authors have incorrectly stated that Dewaele et al. 2018 looked at Crabeater Seals (*Lobodon carcinophagus*) when doing qualitative observations. It was actually a Weddell Seal humerus (*Leoptonychotes weddellii*).

RESPONSE: We have corrected the sentence with the correct species name.

Page 21, Line 2: The recently published paper by Churchill and Uhen (2019) is only briefly discussed here. This paper is a significant advance in the area of phocid taxonomy, and has hard data behind the problems with using isolated postcrania as type specimens. I would like to see a little more nuanced discussion on what they found in terms of the validity of humeri and femora as diagnostic elements.

RESPONSE: Agree, we have included a more extensive discussion on Churchill and Uhen (2019) in page 22.

Page 21, Line 15: “of the taxonomic consequences” instead of “of this taxonomic consequences”.

RESPONSE: We have corrected this sentence.

Page 21, Line 24: “taxonomic diversity, and also” instead of “taxonomic diversity but also”. Alternatively, remove “also might” on Line 23 and replace with “might not only”.

RESPONSE: We agree and we have modified the sentence.

Page 22, Line 1: Remove the second occurrence of “are” (“species are originally”).

RESPONSE: We have corrected the sentence.

Page 22, Line 19-21: Please cite some examples at the end of this sentence.

RESPONSE: We have added a citation.

Page 22, Line 24: “systematic” instead of “systematics”.

RESPONSE: We agree, and we have corrected the sentence.

Page 23, Line 1-3: The example in the brackets needs to be rewritten; as it is wordy and has a few double ups of terminology. Could potentially rewrite as: “(e.g. deposit of fossil specimens in valid natural history collections or research institutions, avoidance of studying specimens derived from illegal or informal trade, as well as from amateur private collectors)”.

RESPONSE: We agree and we have rewritten this sentence.

Page 23, Line 3: Additionally, I find the inclusion of “amateur private collectors” here problematic. In a lot of countries, collections are primarily built by donations from amateur private collectors. Please reword to clarify what you mean (perhaps poachers, or collectors who do not work in conjunction with natural history museums), or consider removing.

RESPONSE: We agree and we have modified this sentence. Now we make reference to amateur private collectors that do not work in conjunction with natural history museums.

Dataset corrections

1. Occurrences of *Callophoca obscura* are completely missing for the Yorktown Formation from Supplementary Table 1, despite being present in Supplementary Table 5. In addition, it is missing from the list of Yorktown Formation pinnipeds on Page 12, Line 6-7.

RESPONSE: With regard to the first point (i.e. occurrences of *Callophoca obscura* are missing from Yorktown Formation from Supp. Table 1), we would like to mention that many of the specimens included in Supp. Table 5 have not been published. However, we did include 3 occurrences of this species from Yorktown Formation that have been published to Supp. Table 1. In addition, we have added the reference of *Callophoca obscura* to the list of species of Yorktown Formation pinnipeds on page 12.

2. A few dates need revising in Supp Table 1:

- *Virginiaphoca magurai* from the Eastover Formation would more specifically originate from the Cobham Bay Member. This is dated to 8.7-6.46 Ma. (Ward and Blackwelder 1980. Stratigraphic revision of upper Miocene and lower Pliocene beds of the Chesapeake group, middle Atlantic Coastal Plain; Dewaele et al. 2018. Diversity of late Neogene Monachinae (Carnivora, Phocidae) from the North Atlantic, with the description of two new species).

RESPONSE: We agree and we have precise the age of this record.

- Keep in mind for the *Properitychus argentinus* records that the paper the PB database references (Cione et al. 2000) for the middle Miocene age (13.82-11.62) of the Parana Formation does not present a strong case for this age. It is more likely Tortonian in age; see Perez et al. 2010. Paleoecological and paleobiogeographic significance of two new species of bivalves in the Parana Formation (late Miocene) of entre rios province Argentina

RESPONSE: We agree and we have corrected the age of this species to the late Miocene.

- The minimum age for *Noriphoca gaudini* is incorrect, it should be 20.44 Ma, not 2.588 Ma. See Dewaele et al 2018 A critical revision of the fossil record, stratigraphy and diversity of the Neogene seal genus *Monotherium* (Carnivora, Phocidae).

RESPONSE: We agree and we have corrected the minimum age of this occurrence.

- Doublecheck the maximum and minimum age records for *Nanophoca vitulinoides*, as sediment samples in Dewaele et al. 2017 have been dated to 14.2-11.63 Ma.

RESPONSE: We have corrected the age of *Nanophoca vitulinoides* to 14.2-12.8 based on the information provided on Dewaele et al. 2017.

- All records for the genus *Frisiphoca* should be conservatively dated to 11.6-5.3 Ma, see Dewaele et al 2018. (A critical revision of the fossil record, stratigraphy and diversity of the Neogene seal genus *Monotherium* (Carnivora, Phocidae)) for a more detailed discussion.

RESPONSE: We agree and we have corrected the age of these occurrences.

- Specimens from the Yorktown Formation may need to be dated to 4.9-3.9 Ma; see discussion of dating in the Supplemental Information of Marx and Fordyce 2015 (Baleen boom and bust: a synthesis of mysticete phylogeny, diversity and disparity).

RESPONSE: We agree and we have modified the age of specimens from Yorktown Formation.

3. A few records from Oceania in Supp Table 1 are erroneous. These are critical to correct given the dearth of records from the Southern Hemisphere:

- No early Miocene phocids were reported from the Port Campbell Limestone of Australia in Fitzgerald 2005.

RESPONSE: We have corrected this record.

- Additionally, phocids from Beaumaris (Australia) reported in Fitzgerald 2004 are the same record as the *Monachinae* from Fordyce and Flannery 1983.

RESPONSE: We have corrected this record.

- The phocids from New Zealand referred to in McKee 1988 are the same phocids from the Tangahoe Formation in McKee 1994. Keep in mind though, the specimens referenced to in these publications (McKee 1988, McKee 1994) are in a private collection, and so no publically accessible specimens from the Tangahoe formation have been published.

RESPONSE: We have corrected these records.

- The Waipunga miroungin specimen from New Zealand (Boessenecker and Churchill 2016) should be dated to 2.4-2.1 Ma, not 3.6-2.588 Ma (the "Late Pliocene" age referred to in the title of this paper is in reference to an older geological timescale, making this an early Pleistocene occurrence).

RESPONSE: We have corrected this data.

4. Double check the records and occurrences for *Properiptychus argentinus* in Supp Table 1, as they may represent the same fossils. Additionally, de Muizon and Bond 1982 in in French, not English.

RESPONSE: We have reviewed these occurrences and we determined that both represent the same specimens. Therefore, we have maintained only the original reference (Ameghino, 1893).

5. The new paper recently published (Velez-Juarbe and Valenzuela-Toro 2019) on two monachine fossils from the Monterey Formation (Tortonian age) in the North Pacific would be useful to include in the dataset. If the analyses need to be re-run, it will be worth including in Supp. Table 1 (provided they have a record in the PB database).

RESPONSE: We have added this reference to Supp. Table 1.